# Single-cell analysis of mosquito hemocytes identifies signatures of immune cell subtypes and cell differentiation

Hyeogsun Kwon[1†], Mubasher Mohammed[2†], Oscar Franzén[3], Johan Ankarklev[2,4], Ryan C Smith[1*]

[1]Department of Entomology, Iowa State University, Ames, United States; [2]Department of Molecular Biosciences, The Wenner-Gren Institute, Stockholm University, Stockholm, Sweden; [3]Integrated Cardio Metabolic Centre, Department of Medicine, Karolinska Institutet, Novum, Huddinge, Sweden; [4]Microbial Single Cell Genomics facility, SciLifeLab, Biomedical Center (BMC) Uppsala University, Uppsala, Sweden

**\*For correspondence:**
smithr@iastate.edu

[†]These authors contributed equally to this work

**Competing interests:** The authors declare that no competing interests exist.

**Abstract** Mosquito immune cells, known as hemocytes, are integral to cellular and humoral responses that limit pathogen survival and mediate immune priming. However, without reliable cell markers and genetic tools, studies of mosquito immune cells have been limited to morphological observations, leaving several aspects of their biology uncharacterized. Here, we use single-cell RNA sequencing (scRNA-seq) to characterize mosquito immune cells, demonstrating an increased complexity to previously defined prohemocyte, oenocytoid, and granulocyte subtypes. Through functional assays relying on phagocytosis, phagocyte depletion, and RNA-FISH experiments, we define markers to accurately distinguish immune cell subtypes and provide evidence for immune cell maturation and differentiation. In addition, gene-silencing experiments demonstrate the importance of lozenge in defining the mosquito oenocytoid cell fate. Together, our scRNA-seq analysis provides an important foundation for future studies of mosquito immune cell biology and a valuable resource for comparative invertebrate immunology.

## Introduction

Across Metazoa, immune cells are vital to promoting wound healing, maintaining homeostasis, and providing anti-pathogen defenses (*Chaplin, 2010*). With immune cells mediating both innate and adaptive immune function in vertebrates, immune cell subtypes display highly specialized roles that have continually been resolved by technological advancements that enable their study (*Papalexi and Satija, 2018*; *Proserpio and Mahata, 2016*). Recently, the advent of single-cell sequencing (scRNA-seq) has continued to delineate and provide further resolution into new cell types and immune cell functions in mammals (*Szabo et al., 2019*; *Villani et al., 2017*). In lesser studied invertebrates lacking adaptive immunity, single-cell technologies have enhanced descriptions of previously described cell types and have redefined cell complexity (*Cattenoz et al., 2020*; *Cho et al., 2020*; *Raddi et al., 2020*; *Severo et al., 2018*; *Tattikota et al., 2020*).

In insects, hematopoiesis and immune cell function have predominantly been examined in Lepidoptera and *Drosophila* (*Banerjee et al., 2019*; *Lavine and Strand, 2002*), with the mosquito, *Anopheles gambiae*, recently serving as an emerging study system (*Kwon and Smith, 2019*). Mosquito immune cells (hemocytes) have proven integral to the cellular and humoral responses that limit invading pathogens in the mosquito host (*Baton et al., 2009*; *Castillo et al., 2017*; *Hillyer et al.,*

*2003a*; *Kwon and Smith, 2019*; *Smith et al., 2016*) and the establishment of immune memory (*Ramirez et al., 2015*; *Rodrigues et al., 2010*). Transcriptional (*Baton et al., 2009*; *Pinto et al., 2009*) and proteomic (*Smith et al., 2016*) analysis of mosquito hemocyte populations have yielded important information into the regulation of hemocyte function in response to blood-feeding and infection. However, the study of mosquito immune cells has been complicated by discrepancies in cell classification (*Ribeiro and Brehélin, 2006*), cell numbers (*Hillyer and Strand, 2014*), and methodologies to examine their function (*Kwon and Smith, 2019*). These constraints are magnified by the lack of genetic tools and markers that have limited studies of immune cells outside of *Drosophila* strictly to morphological properties of size and shape (*Hillyer and Strand, 2014*; *Kwon and Smith, 2019*).

Traditional classifications of mosquito hemocytes describe three cell types: prohemocyte precursors, phagocytic granulocytes, and oenocytoids that have primary roles in melanization (*Castillo et al., 2006*). However, recent studies have begun to challenge these traditional immune cell classifications, demonstrating the existence of multiple types of phagocytic cells (*Kwon and Smith, 2019*; *Severo et al., 2018*) and that both granulocytes and oenocytoids contribute to prophenoloxidase expression (*Bryant and Michel, 2016*; *Kwon et al., 2020*; *Kwon and Smith, 2019*; *Severo et al., 2018*; *Smith et al., 2016*). Together, this suggests that there is additional complexity to mosquito immune cells that are not accurately represented by the traditional classification of mosquito hemocytes into three cell types.

For this reason, here we employ the use of scRNA-seq using a Smart-seq2 methodology to generate full-length sequence coverage to better characterize mosquito immune cell populations. Using a conservative approach, we identify seven hemocyte subtypes with distinct molecular signatures and validate these characterizations using a variety of bioinformatic and experimental molecular techniques. We define new markers that can accurately distinguish immune cell subtypes, improving upon the ambiguity of existing methodologies. Moreover, our data support a new model of immune cell differentiation and maturation that leads to a dynamic population of circulating immune cells in the adult female mosquito. In summary, these data represent a valuable resource to advance the study of mosquito immune cells, offering a robust data set for comparative immunology with other insect systems.

## Results

### Isolation of mosquito immune cells and scRNA-seq analysis

To examine mosquito immune cells by scRNA-seq, adult female *An. gambiae* were perfused as previously (*Kwon et al., 2017*; *Kwon and Smith, 2019*; *Reynolds et al., 2020*; *Smith et al., 2016*; *Smith et al., 2015*) from either naive or blood-fed (24 hr post-feeding) conditions to assess the physiological impacts of blood-feeding on hemocyte populations as previously suggested (*Bryant and Michel, 2016*; *Bryant and Michel, 2014*; *Castillo et al., 2011*; *Reynolds et al., 2020*). Following perfusion, cells were stained with a live-dead viability stain to select for live cells, with mosquito immune cells distinguished by labeling with FITC-conjugated wheat germ agglutinin (WGA) as a general hemocyte marker and the far-red stain DRAQ5 to label DNA content as previously (*Kwon and Smith, 2019*). Based on consistent patterns of WGA/DRAQ5 signal intensity that were suggestive that these labeling properties could distinguish distinct groups of immune cells (*Figure 1A*, *Figure 1—figure supplement 1*), we isolated individual cells by fluorescence-activated cell sorting (FACS) using three 'gates' to enrich for defined cell populations using these WGA/DRAQ5 properties (*Figure 1A*, *Figure 1—figure supplement 1*). An additional, non-selective fourth gate isolated cells at random to achieve an unbiased cell population that would be influenced by overall cell abundance (*Figure 1A*). Based on these parameters, individual cells were isolated by FACS into a 384-well plate for further processing for scRNA-seq using the SMART-seq2 methodology (*Picelli et al., 2014*; *Picelli et al., 2013*). A total of 262 cells passed the quality filtering threshold of 10,000 reads per cell (*Figure 1—figure supplement 2*), yielding 194 and 68 cells, respectively from naive and blood-fed conditions (*Supplementary file 1*). Overall, we detected expression (>0.1 RPKM) from ~46% (6352/13,764) of the *An. gambiae* genome, with a median of 1646 genes expressed per cell (range 45–5215), comparable to *Severo et al., 2018*. However, our data display a higher number of genes per cell and larger variance in genes between cell types, patterns suggestive of a broader

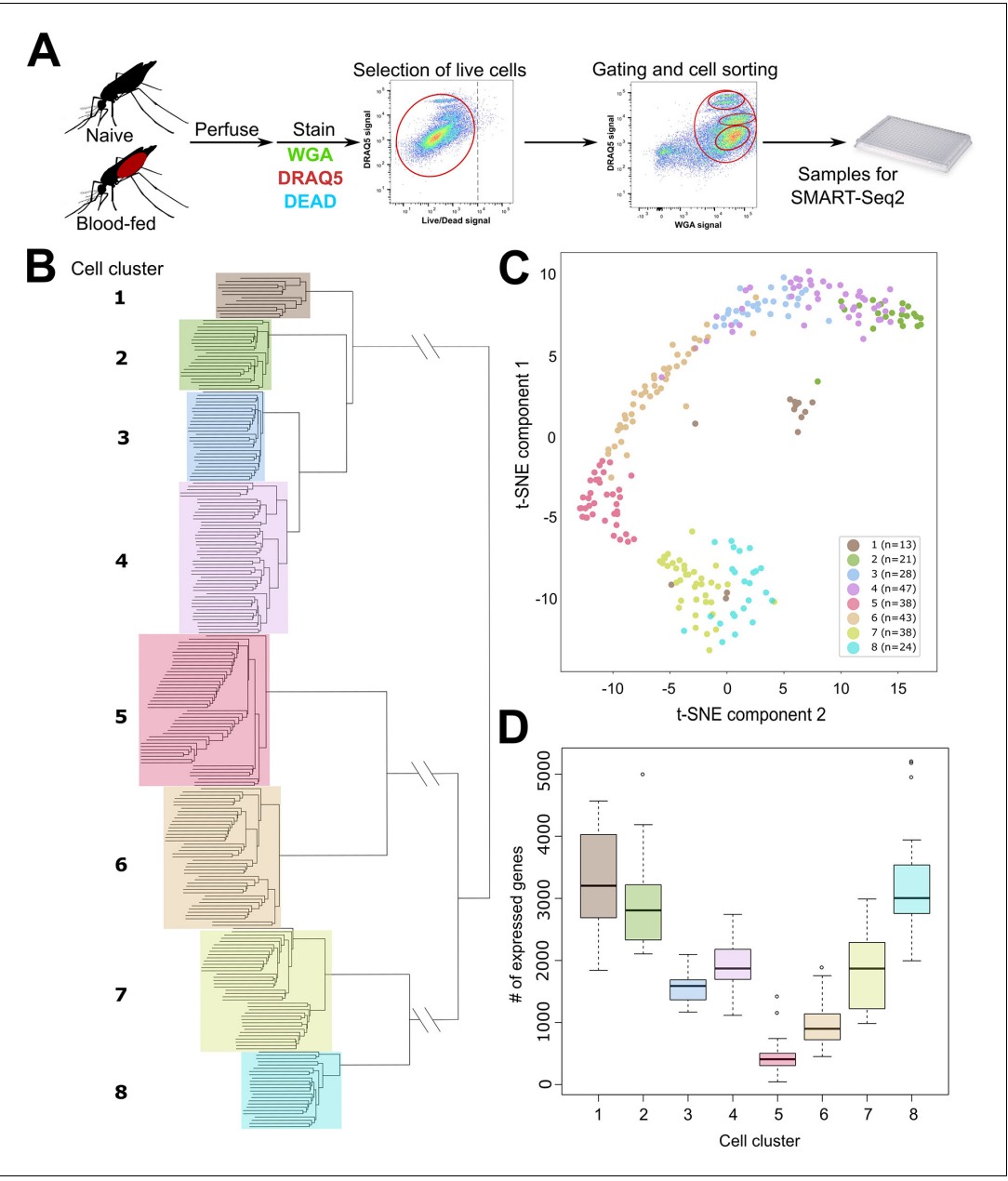

**Figure 1.** scRNA-seq of *An. gambiae* immune cells. (**A**) Graphical overview of the isolation of mosquito immune cells from naive and blood-fed mosquitoes. Following perfusion, cells were stained to enable processing by fluorescent activated cell sorting (FACS) and isolation for scRNA-seq. Resulting immune cells data were separated into eight-cell clusters based on hierarchical clustering analysis (**B**) and visualized using a t-Distributed Stochastic Neighbor Embedding (t-SNE) plot (**C**). The number of expressed genes per cluster are displayed as a boxplot (**D**). The online version of this article includes the following figure supplement(s) for figure 1:

**Figure supplement 1.** FACS sorting of mosquito immune cell populations.

**Figure supplement 2.** Quality assessment of scRNA-seq analysis on mosquito immune cells.

**Figure supplement 3.** Naive and blood-fed immune cells isolated in FACS analysis.

**Figure supplement 4.** Examination of differentially regulated genes in Clusters 2 and 4 under naive and blood-fed conditions.

range of cell populations represented in our dataset. All immune cell data can be visualized and searched using the following database: https://alona.panglaodb.se/results.html?job= 2c2r1NM5Zl2qcW44RSrjkHf3Oyv51y_5f09d74b770c9.

Using hierarchical clustering, we conservatively define eight distinct cell clusters or immune cell-subtypes (*Figure 1B*). These clusters are supported by the unique molecular profiles of each cell cluster when analyzed by tSNE (*Figure 1C*), as well as variability in the number of expressed genes (*Figure 1D*) that infers functional heterogeneity in these cell populations. When referenced to our FACS gating methodology based on WGA/DRAQ5 staining (*Figure 1A*, *Figure 1—figure supplement 1*), each cell cluster is represented in our targeted, yet all-inclusive gating conditions (Gate 4), although these gating conditions (Gate 4) were only performed under naïve conditions (*Figure 1— figure supplement 1*). Moreover, each of the specific gating conditions (Gates 1–3) provides enrichment for distinct cell types under both naïve and blood-fed conditions (*Figure 1—figure supplement 1*). Clusters 1, 7, and 8 are enriched in Gate 1, Clusters 5 and 6 in Gate 2, and Clusters 2–4 in Gate 3 (*Figure 1—figure supplement 1*), arguing that these cells have similar physical properties in WGA staining and DNA content. While at present, it is unclear what defines these properties on the molecular level, the enrichment achieved through our gating strategy provides further support for our FACS methodology. The differences in DNA content amongst the FACS gating conditions are suggestive of potential variations in ploidy (the number of sets of chromosomes) in hemocyte populations as previously suggested (*Bryant and Michel, 2016*; *Bryant and Michel, 2014*), although ploidy levels for individual hemocyte subtypes have not previously been described. As a result, our data suggest that differences in DNA content can be further utilized to distinguish immune cell subtypes (*Figure 1—figure supplement 1*).

With the exception of Cluster 3, which was only identified in naïve mosquitoes, each of the respective cell clusters were found under both naïve and blood-fed conditions (*Figure 1—figure supplement 3*). When paired with differential gene expression between naive and blood-fed cells of each cluster, only Clusters 2 and 4 displayed significant changes in gene expression (*Supplementary file 2*). Interestingly, this includes the down-regulation of several immune genes (TEP1, SCRASP1, and LYSC1) following blood-feeding in Cluster 2 that have previously defined roles in pathogen defense (*Blandin et al., 2004*; *Kajla et al., 2011*; *Levashina et al., 2001*; *Smith et al., 2016*), while genes involved in vesicle trafficking (Rab6A) and redox metabolism display increased expression (*Figure 1—figure supplement 4*, *Supplementary file 2*). Cluster 4 displayed an increase in gene expression for a serrate RNA effector molecule (Ars2), an integrin (AGAP006826), and a Sp2 transcription factor (AGAP004438) following blood-feeding, while an enzyme involved in the processing of glycoproteins (AGAP000249) was significantly reduced (*Figure 1—figure supplement 4*, *Supplementary file 2*). Together, these data suggest that blood-feeding may primarily influence the activation state and gene expression of specific immune cell subtypes as previously suggested (*Bryant and Michel, 2016*; *Bryant and Michel, 2014*; *Reynolds et al., 2020*; *Smith et al., 2016*).

## Characterization of *An. gambiae* immune cell clusters

To further characterize the cell clusters resulting from our scRNA-seq analysis, we used the Seurat package (*Butler et al., 2018*) to identify transcriptional markers significantly enriched for each cell cluster (*Figure 2A*, *Supplementary file 3*). When the expression of mitotic markers (*Raddi et al., 2020*) was examined under naive (sugar-fed) and blood-fed conditions, no discernable differences were detected between physiological conditions (*Figure 2B*). The expression of mitotic markers was also compared across individual cell clusters, enabling comparisons between naive and blood-fed conditions (*Figure 2C*). Clusters 2 and 4 displayed the highest expression of mitotic markers (*Figure 2C*), supporting that these cells may have some capacity for proliferation. Blood-feeding impacted these populations inversely, with increased marker expression in Cluster 2 and a decrease in Cluster 4; however, these results were not significant (*Figure 2C*). To more closely evaluate the molecular profiles of each cell cluster, transcripts identified in more than 80% of cells in each cluster (*Supplementary file 4*) were used to perform gene ontology (GO) analysis (*Figure 2D*). Comparisons across cell clusters provide further support that Clusters 2–4 are highly analogous in their core machinery, with Cluster 6 displaying a related, yet divergent cellular composition represented by an increased representation of transcripts involved in translation (*Figure 2D*). Correlations with a previous proteomics study of phagocytic granulocytes in *An. gambiae* (*Smith et al., 2016*) demonstrate that transcripts of Clusters 2–4 have the strongest associations with phagocytic immune cells

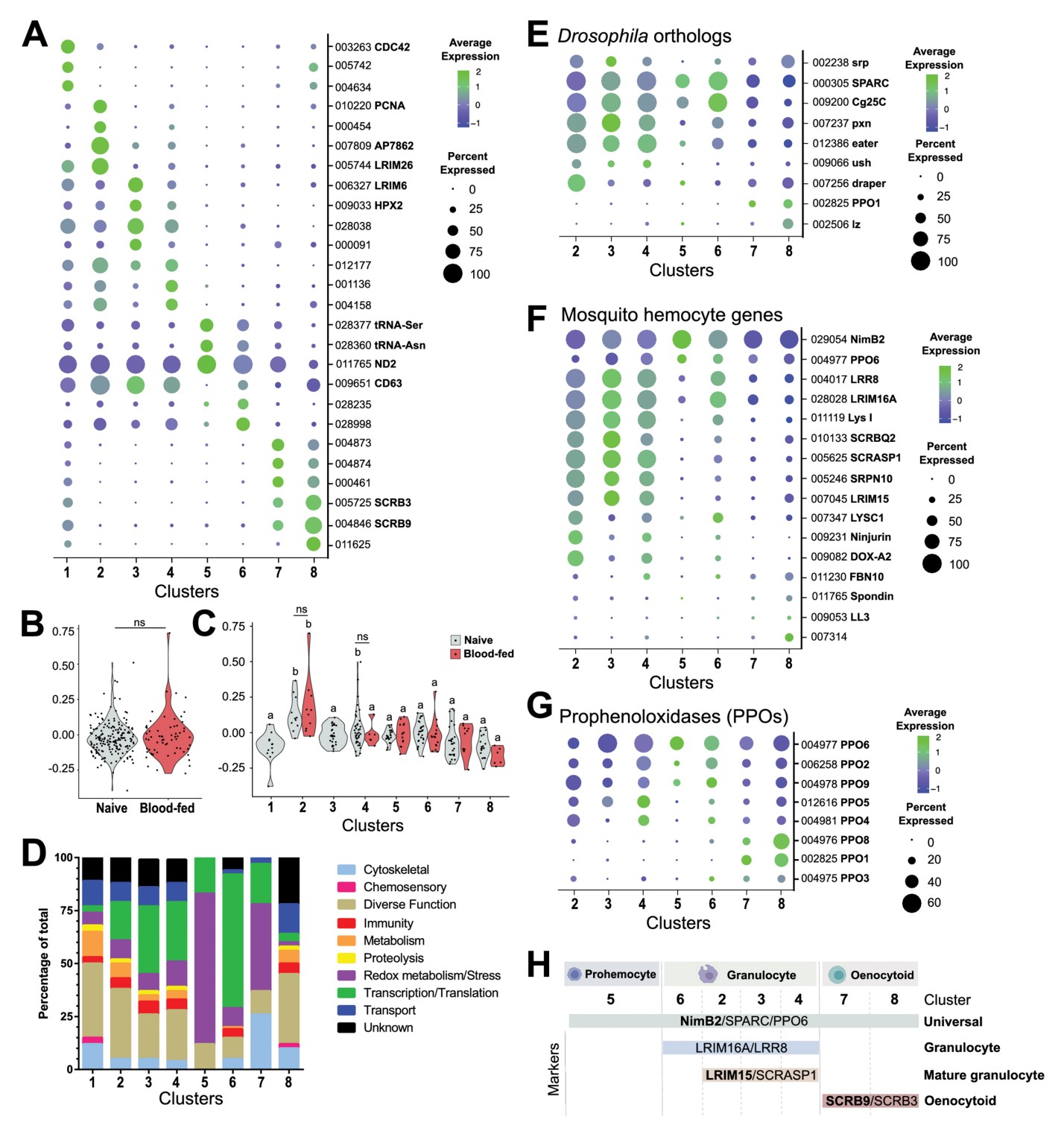

**Figure 2.** Comparative analysis of mosquito immune cells. (**A**) Marker gene expression displayed by dot plot across cell clusters. Dot color shows levels of average expression, while dot size represents the percentage of cells expressing the corresponding genes in each cell cluster. (**B**) Violin plot of cell cycle genes (GO::0007049) displayed as the difference in average gene expression levels between the cell cycle gene set in cells under naïve and blood-fed conditions. (**C**) Similar comparisons of cell cycle genes were examined in individual cell clusters and under naïve and blood-fed conditions where possible. For **B** and **C**, positive numbers indicate higher levels of cell cycle gene expression compared to the random set for that physiological condition or cell cluster. (**D**) Gene ontology (GO) analysis of genes expressed in >80% cells within each respective cluster. Heat maps of candidate

*Figure 2 continued on next page*

*Figure 2 continued*

genes to *Drosophila* hemocyte orthologs (**E**), described mosquito hemocyte genes (**F**), or *An. gambiae* prophenoloxidases (PPOs) (**G**) to enable the characterization of immune cells form each cell cluster. (**H**) From these analysis, immune cells cluster were assigned to tentative cell types (prohemocytes, granulocytes, oenocytoids) based on the expression subtype-specific marker expression. Genes in bold are featured prominently in our downstream analysis.

The online version of this article includes the following figure supplement(s) for figure 2:

**Figure supplement 1.** Comparisons of immune cell clusters to the *An.*
**Figure supplement 2.** Comparisons of Cluster 1 to non-hemocyte cell populations.
**Figure supplement 3.** Expression of hemocyte genes across all cell clusters.
**Figure supplement 4.** Expression of mosquito immune genes across cell clusters.
**Figure supplement 5.** Expression of serine protease inhibitors (SRPNs) and CLIP serine proteases across cell clusters.
**Figure supplement 6.** Expression of chemosensory genes across cell clusters.
**Figure supplement 7.** Expression of specific tRNAs across cell clusters.
**Figure supplement 8.** Candidate markers of immune cell clusters and hemocyte sub-types.

(*Figure 2—figure supplement 1*), providing support that these clusters represent populations of phagocytic granulocytes. Cells in Cluster 5 display a unique profile predominantly comprised of genes implicated in redox metabolism/stress responses (*Figure 2D*), while Clusters 7 and 8 display marked differences in composition (*Figure 2D*), despite sharing similar markers to delineate these cell types (*Figure 2A*).

Cells in Cluster 1 display an increased representation of genes involved in metabolic function and a decrease in genes involved in transcription and translation (*Figure 2D*), that when paired with the generalized expression of otherwise cluster-specific markers (*Figure 2A*, *Figure 2—figure supplement 2*) suggest that the cells within this cluster are distinct from other cell types in our analysis. Since hemolymph perfusion (to isolate circulating hemocytes) can often be contaminated by fat body cells or other cellular debris (*Figure 2—figure supplement 2*; *Castillo et al., 2006*; *Smith et al., 2016*), and have been identified as contaminants in other hemocyte single-cell studies (*Raddi et al., 2020*; *Tattikota et al., 2020*), we examined the possibility that Cluster 1 may represent non-hemocyte contaminating cells. Using the 'fat body' and 'muscle' enriched gene sets for *An. gambiae* defined by *Raddi et al., 2020* and the 'non-hemocyte' gene set in *Drosophila* from *Tattikota et al., 2020* for comparison to our cell clusters, we demonstrate that genes expressed in Cluster 1 closely resemble the profiles of non-hemocyte cell types and likely represents cellular debris (such as fat body or oenocytes) associated with perfusion techniques (*Figure 2—figure supplement 2*). This is further supported by the high DNA content of Cluster 1 cells identified by our FACS methodology (*Figure 1—figure supplement 1*), where previous studies in mosquitoes (*Dittmann et al., 1989*) and other insect species (*Ren et al., 2020*) have demonstrated that fat body cells display increased levels of cell ploidy. Alternatively, Cluster 1 may also represent cell doublets of mixed cell origins (fat body, granulocytes, or oenocytoids) resulting from errors in our FACS isolation methodology. This is supported by the expression of several markers (such as *LRIM26* and *SCRB9*) at high levels that otherwise define specific cell clusters (*Figure 2A*) and the expression of known *Drosophila* and mosquito hemocyte genes (*Figure 2—figure supplement 3*). While Cluster 1 may also represent cell types undergoing transdifferentiation as has been suggested in *Drosophila* (*Leitão and Sucena, 2015*), the absence of increased levels of cell cycle genes (*Figure 2C*) previously implicated in dividing hemocytes (*Raddi et al., 2020*), argues that this is a less likely scenario. Additional considerations that these cells are pluripotent precursors or represent recently described megacyte populations (*Raddi et al., 2020*) also seem unlikely given that pluripotent precursors have not been described in other insect single-cell studies (*Cattenoz et al., 2020*; *Cho et al., 2020*; *Raddi et al., 2020*; *Severo et al., 2018*; *Tattikota et al., 2020*) and that Cluster 1 cells do not display enriched expression of *TM7318* and *LL3* (*Supplementary file 5*) that are indicative of megacytes (*Raddi et al., 2020*). Taken together, without a well-defined expression pattern and the potential that these cells may be experimental artifacts, cells of Cluster 1 were not included in the further downstream analysis of our immune cell populations.

In order to further determine the immune cell classifications of our remaining clusters, we examined the expression of well-characterized *Drosophila* hemocyte gene orthologs (*Dudzic et al., 2015*; *Evans et al., 2014*; *Fossett et al., 2003*; *Franc et al., 1996*; *Kocks et al., 2005*; *Manaka et al.,*

*2004*; *Martinek et al., 2011*; *Stofanko et al., 2008*; *Waltzer et al., 2003*) in our dataset (*Figure 2E*). SPARC and Cg25C were expressed at high levels across cell clusters, suggesting that these could be universally expressed markers of mosquito immune cells (*Figure 2E*). Clusters 2–4 and 6 express the *Drosophila* plasmatocyte markers (equivalent to mosquito granulocytes) *peroxidasin* (pxn) and *eater* (*Figure 2E*) suggestive of phagocytic cell function. In contrast, the expression of *lozenge* (lz) and *PPO1* indicative of *Drosophila* crystal cells (*Cattenoz et al., 2020*; *Dudzic et al., 2015*; *Evans et al., 2014*; *Fossett et al., 2003*; *Tattikota et al., 2020*) were most prevalent in Clusters 7 and 8 (*Figure 2E*), suggesting that these cell clusters are representative of mosquito oenocytoid populations (equivalent to crystal cells). However, little resolution into the role of Cluster 5 was provided through these comparisons to known *Drosophila* markers (*Figure 2E*).

Similar classifications were performed for described mosquito hemocyte genes (*Bryant and Michel, 2016*; *Castillo et al., 2006*; *Danielli et al., 2000*; *Estévez-Lao and Hillyer, 2014*; *Kwon and Smith, 2019*; *Midega et al., 2013*; *Pinto et al., 2009*; *Raddi et al., 2020*; *Severo et al., 2018*; *Smith et al., 2015*; *Smith et al., 2016*) to provide additional resolution into our mosquito immune cell clusters (*Figure 2F*). As previously suggested (*Kwon and Smith, 2019*), the expression of NimB2 and PPO6 support their use as universal markers of mosquito immune cells (*Figure 2F*). Through the use of described granulocyte markers (*Danielli et al., 2003*; *Raddi et al., 2020*; *Smith et al., 2016*), we are able to describe two discrete phagocytic cell types (*Figure 2F*). The expression of *LRIM16A*, *LRR8*, and *LYS I* in cells of Clusters 2–4 and 6 suggest that these cells are granulocytes in origin (*Raddi et al., 2020*; *Severo et al., 2018*; *Smith et al., 2016*), while the expression of *SCRASP1*, *SRPN10*, and *LRIM15* (*Danielli et al., 2003*; *Smith et al., 2016*) in Clusters 2–4 suggest that these are more specialized granulocyte populations (*Figure 2F*). Other markers such as *Ninjurin* (*Pinto et al., 2009*) and *DOX-A2* (*Castillo et al., 2006*) expressed in Clusters 2 and 4 further delineate these presumed phagocytic cell types (*Figure 2F*). In addition, *AGAP007314* grouped strongly with the presumed oenocytoid cell population of Cluster 8, supporting its previously described roles in melanization (*Pinto et al., 2009*).

When we examine the transcriptional profiles of prophenoloxidases (PPOs), a family of enzymes that catalyze the production of melanin in response to infection (*Dudzic et al., 2015*), we demonstrate that the eight PPOs detected in our analysis are expressed in each of the major immune cell subtypes (*Figure 2G*). As previously suggested (*Kwon and Smith, 2019*; *Severo et al., 2018*), PPO6 is universally expressed in all hemocytes (*Figure 2F and G*). *PPO2*, *PPO4*, *PPO5*, and *PPO9* are most abundant in putative granulocyte populations, while *PPO1*, *PPO3*, and *PPO8* are enriched in putative oenocytoids (*Figure 2G*). This directly contrasts previous suggestions that mosquito PPOs are only constitutively expressed in oenocytoid populations (*Castillo et al., 2006*; *Hillyer and Strand, 2014*; *Strand, 2008*), yet is supported by recent evidence that phagocytic granulocyte populations in mosquitoes significantly contribute to *PPO* production (*Kwon and Smith, 2019*; *Smith et al., 2016*). Furthermore, the enriched expression of *PPO1*, *PPO8*, and (to a lesser extent) *PPO3* in oenocytoids is supported by recent studies examining prostaglandin signaling on *PPO* expression in mosquito oenocytoid populations (*Kwon et al., 2020*).

Additional characterizations of immune signaling pathways (*Figure 2—figure supplement 4*), SRPNs and CLIPs (*Figure 2—figure supplement 5*), chemosensory receptors/proteins (*Figure 2—figure supplement 6*), and tRNA expression (*Figure 2—figure supplement 7*) across cell clusters provide further detail into the functions of our tentative immune cell clusters. We demonstrate that known anti-microbial genes and signaling components of the Toll, IMD, and JAK-STAT pathways (*Cirimotich et al., 2010*) display the highest expression in Clusters 2 and 8 (*Figure 2—figure supplement 4*), similar to the expression patterns of SRPNs and CLIPs (*Figure 2—figure supplement 5*) that mediate immune activation (*Gulley et al., 2013*; *Kanost and Jiang, 2015*). This suggests that mosquito granulocyte and oenocytoid populations each contribute to the expression of a distinct subset of immune signaling processes. However, at present, it is unclear if this corresponds to pathogen-specific defenses or immune responses unique to each particular cell type. Interestingly, receptors involved in chemosensory recognition (ionotropic, gustatory, and odorant receptors; odorant binding proteins) are highly expressed in Clusters 7 and 8 (*Figure 2—figure supplement 6*). Although their function has not been described in mosquitoes, the role of odorant binding proteins on immune system development has been described previously in other insect systems (*Benoit et al., 2017*). Moreover, the differential expression of transfer RNA (tRNA) genes across cell populations provided useful measures to tease apart Clusters 5–8 from other cell clusters (*Figure 2—*

*figure supplement 7*), potentially representing different activation states or stages of immune cell development as previously defined in mammalian systems (*Krishna et al., 2019*; *Rak et al., 2020*; *Torrent et al., 2018*).

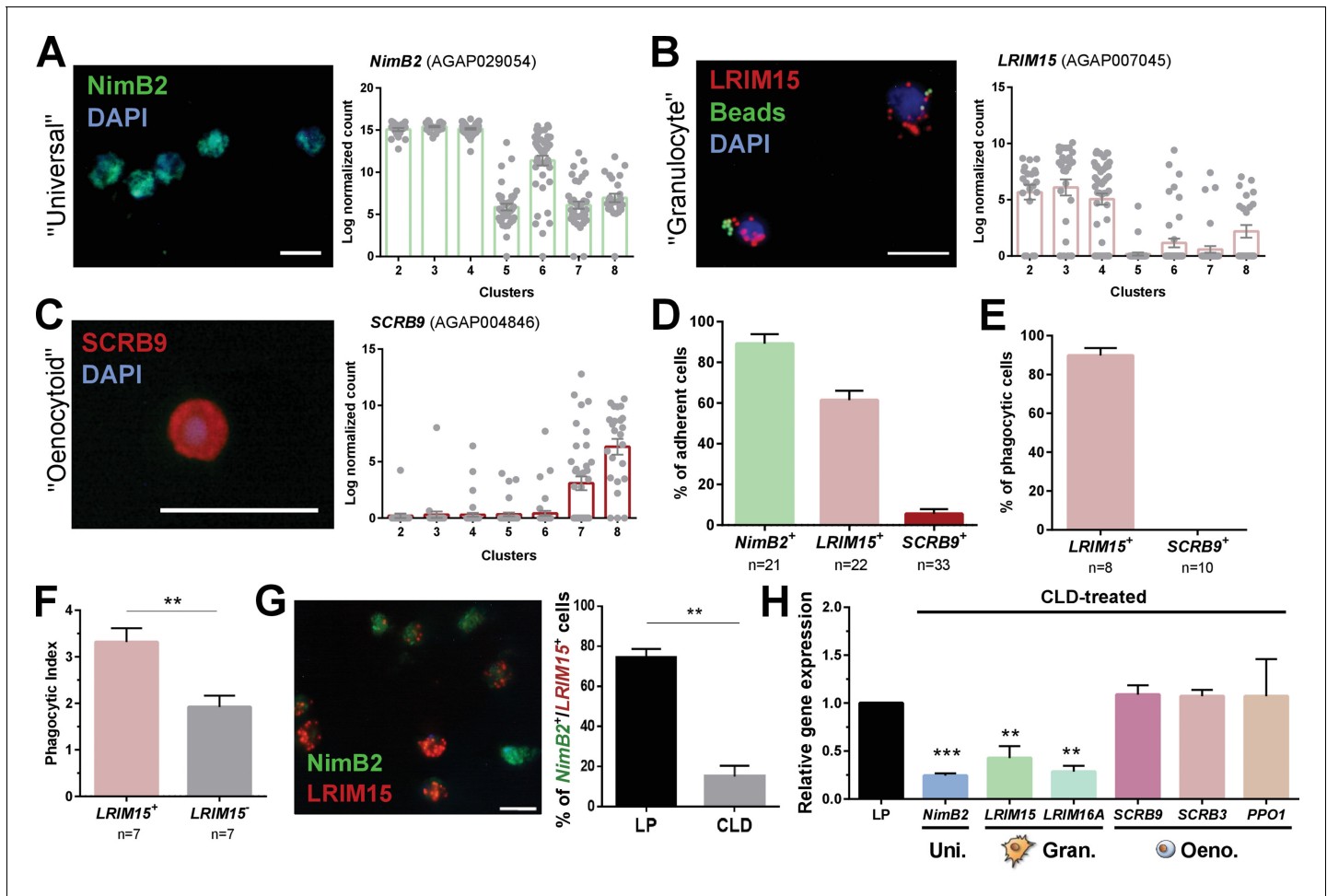

**Figure 3.** Definition of mosquito immune cell subtypes. RNA-FISH and gene expression profiles across cell clusters for the 'universal' marker, *NimB2* (A), the 'granulocyte' marker, *LRIM15* (B), and 'oenocytoid' marker, *SCRB9* (C). The percentage of adherent cells following fixation was evaluated for each of the respective *NimB2*, *LRIM15*, and *SCRB9* markers in four or more independent replicates (D). To determine the phagocytic ability of granulocytes and oenocytoids, the uptake of fluorescent beads was evaluated in either LRIM15[+] or SCRB9[+] cells in two independent replicates (E). LRIM15[+] cells display a higher phagocytic index (# beads engulfed per cell) than LRIM15[-] cell populations (F). Data were analyzed using a Mann–Whitney test, with bars representing mean ± SE of two independent replicates. The phagocytic ability of LRIM15[+] cells was further validated by examining the abundance of NimB2[+]/LRIM15[+] cells by RNA-FISH following perfusion after treatment with control (LP)- or clodronate (CLD) liposomes that deplete phagocytic cells (G). Data were analyzed using a Mann–Whitney test. Bars represent mean ± SE of two independent replicates. Additional validation of clodronate (CLD) depletion of phagocytic cells was performed by qRT-PCR using primers for universal (uni.), granulocyte (gran.), and oenocytoid (oeno.) cell markers. Data were analyzed using an unpaired t test to determine differences in relative gene expression between LP and CLD treatments. Bars represent mean ± SE of three independent replications (H). Asterisks denote significance (**p < 0.01, ***p < 0.001). Scale bar, 10 μm. The online version of this article includes the following source data and figure supplement(s) for figure 3:

**Source data 1.** Source data pertaining to data presented in *Figure 3D–H*.
**Figure supplement 1.** RNA-FISH of *SCRB9*[+] immune cells.
**Figure supplement 2.** Expression profile and RNA-FISH of *SCRB3*[+] immune cells.
**Figure supplement 3.** Co-hybridization of SCRB9 and LRIM15 labels distinct immune cell populations by RNA-FISH.
**Figure supplement 4.** Expression of granulocyte and oenocytoid markers to distinguish immune cell sub-types.
**Figure supplement 5.** Examination of conserved hemocyte markers across *Anopheles* and *Drosophila* single-cell studies.
**Figure supplement 6.** Examination of mosquito hemocyte markers in *Anopheles* single-cell studies.
**Figure supplement 7.** Comparison of immune cell clusters to Raddi et al.
**Figure supplement 8.** Comparisons of immune cell clusters to previously defined PPO6[low] and PPO6[high] hemocyte populations.

Based on these characterizations, our data support the identification of NimB2, SPARC, and PPO6 as universal markers of mosquito immune cell populations that can be found in each of our cell clusters (*Figure 2H*, *Figure 2—figure supplement 8*). Given the low number of expressed genes (*Figure 1*), the lack of discernable markers (*Figure 2*, *Figure 2—figure supplement 8*), and low levels of cyclin G2 (*Supplementary file 5*) that define differentiated cell populations (*Horne et al., 1997*; *Martínez-Gac et al., 2004*), we believe that Cluster 5 represents a progenitor population of prohemocytes (*Figure 2H*). Moreover, the high expression of NimB2 and SPARC (*Figure 2E and F*) in Cluster 5 cells are comparable to the less differentiated prohemocyte populations described by *Raddi et al., 2020*. Clusters 2–4 and 6 can be described as granulocytes, distinguished by LRIM16A and LRR8 (*Raddi et al., 2020*; *Smith et al., 2016*), and further delineated as 'mature' granulocyte populations in Clusters 2–4 marked by the expression of LRIM15 and SCRASP1 (*Figure 2H*, *Figure 2—figure supplement 8*). In the absence of these additional phagocytic markers, we believe that the less defined populations of Cluster 6 likely represent immature granulocytes. Clusters 7 and 8 represent populations of oenocytoids that can easily be denoted by the expression of two scavenger receptors, SCRB9 and SCRB3 (*Figure 2H*, *Figure 2—figure supplement 8*).

## Confirmation of mosquito immune cell subtypes

To confirm the identification of our immune cell clusters and to establish a reliable set of markers to distinguish immune cell subtypes, we performed RNA-FISH on fixed immune cell populations and paired these observations with the phagocytic properties of each of the respective cell populations (*Figure 3*). Supported by our expression data, the 'universal' marker *Nimrod B2* (NimB2) labeled all hemocytes (*Figure 3A*). Serving as a marker for phagocytic cells, we demonstrate that *LRIM15* effectively labels phagocytic cell populations (*Figure 3B*), while the labeling of *SCRB9* (*Figure 3C*, *Figure 3—figure supplement 1*) and *SCRB3* denote mosquito oenocytoid populations (*Figure 3—figure supplement 2*). As further validation, RNA-FISH experiments performed with both *LRIM15* and *SCRB9* probes identify distinct populations of *LRIM15*[+] or *SCRB9*[+] cells, confirming that these markers label unique cell populations (*Figure 3—figure supplement 3*).

When these respective RNA-FISH markers are used to examine cell abundance, >90% of fixed cells are *NimB2*[+] (*Figure 3D*), further demonstrating its role as a reliable general cell marker. *LRIM15*[+] phagocytic granulocyte populations represent ~60% of fixed cells, while only ~5% of cells are *SCRB9*[+] (*Figure 3D*). As expected, *LRIM15*[+] cells are phagocytic, while *SCRB9*[+] cells do not display phagocytic activity (*Figure 3E*), agreeing with the respective phagocytic and non-phagocytic roles of mosquito granulocytes and oenocytoids. Moreover, when phagocytic activity was compared between *LRIM15*[+] and *LRIM15*[-] phagocytes, *LRIM15*[+] cells displayed significantly higher phagocytic activity (*Figure 3F*). These data also indirectly support that the *LRIM15*[-] phagocytic cells are likely those of the 'immature' granulocytes of Cluster 6. Additional experiments using clodronate liposomes to deplete phagocytic cell populations (*Kwon and Smith, 2019*) demonstrate that *NimB2*[+]/ *LRIM15*[+] cells are highly susceptible to clodronate treatment (*Figure 3G*), providing further confirmation of their phagocytic cell function. The specificity of clodronate treatment was further validated by qRT-PCR, demonstrating that the expression of 'universal' or 'granulocyte' transcripts were reduced following phagocyte depletion, while 'oenocytoid' markers remain unaffected (*Figure 3H*). Together, these data confirm the identification of our mosquito immune cell clusters and define the use of specific cell markers to delineate granulocyte and oenocytoid populations in *An. gambiae*.

However, ~30% of cells (most are *NimB2*[+]) cannot be fully resolved by the expression of *LRIM15* and *SCRB9* alone (*Figure 3B and C*). This is evidenced by the *NimB2*[+]/*LRIM15*[-] cells displayed in *Figure 3F*, which may represent other adherent cell populations with 'granulocyte-like' morphology, potentially belonging to cells of Clusters 5 or 6 that are *LRIM15*[-] (*Figure 3B*).

## Further defining mosquito granulocyte and oenocytoid sub-populations

Based on the initial comparisons of known phagocytic cell markers (*Figure 2*) and the confirmation of phagocytic activity in *LRIM15*[+] cells (*Figure 3*), our data conservatively support the presence of four granulocyte subtypes (Clusters 2, 3, 4, and 6; *Figure 2*, *Figure 2—figure supplement 7*). To better define these subtypes, we more closely investigated the putative functional roles of each of these cell populations. Cells of Cluster two display high levels of immune gene expression of antimicrobial peptides (AMPs), components of the Toll pathway, *TEP1*, *MMP1*, and *LRIM26* (*Figure 3—*

*figure supplement 4*) that likely represent a class of specialized immune cells similar to other recent studies (*Cattenoz et al., 2020*; *Raddi et al., 2020*; *Tattikota et al., 2020*). However, tentative cell functions for the other cell clusters are less transparent. Cells of Cluster three have little immune gene expression and are distinguished by the increased production of *LRIM6*, *cathepsin L*, and *cathepsin F*, while cells in Cluster four display high levels of *FBN 8*, *FBN 10*, and multiple PPO genes (*Figure 3—figure supplement 4*). By contrast, cells in Cluster 6 display reduced expression of several phagocytic markers (*Figure 3—figure supplement 4*), suggesting that cells of this subtype lack the specialized phagocytic function of fully differentiated granulocytes. The reduced expression of *Cyclin G2* (*Figure 3—figure supplement 4*), a marker of differentiated cells (*Horne et al., 1997*; *Martínez-Gac et al., 2004*), supports this hypothesis. Together, these data support that Cluster 6 likely represents a granulocyte precursor, whereas Clusters 2–4 are differentiated subtypes with unique cell functions.

Similarly, differences in gene expression were also used to distinguish between the two oenocytoid subtypes. While both cell clusters express a subset of genes unique to the oenocytoid lineage (*Figure 2*), expression of most oenocytoid markers, such as *PPO1*, *SCRB3* and *SCRB9*, are higher in Cluster 8 than in Cluster 7 (*Figure 3—figure supplement 4*). Moreover, Cluster 8 also expresses high levels of *hnt/peb*, *DnaJ-1*, *Mlf*, *klu*, and *lozenge* (*Figure 3—figure supplement 4*) that are indicative of mature *Drosophila* crystal cells (*Koranteng et al., 2020*; *Miller et al., 2017*; *Tattikota et al., 2020*; *Terriente-Felix et al., 2013*), suggesting that Clusters 7 and 8 respectively represent populations of immature and mature oenocytoids similar to comparable populations of crystal cells in *Drosophila* (*Cho et al., 2020*; *Tattikota et al., 2020*). Interestingly, the isolation of these oenocytoid populations predominantly in Gate 1 (*Figure 1—figure supplement 1*) of our FACS methodology suggest that these cells may be polyploid, which may enhance their ability to rapidly undergo protein synthesis in response to immune challenge as previously proposed (*Ren et al., 2020*).

## Comparative analysis to other hemocyte single-cell studies in flies and mosquitoes

When we compare these cell identifications to other hemocyte single-cell studies in *Drosophila* (*Tattikota et al., 2020*) and *An. gambiae* (*Raddi et al., 2020*), we see both similarities and differences between these studies and our own. Using markers conserved across insect systems, we demonstrate that *NimB2* and *SPARC* represent excellent universal hemocyte markers (*Figure 3—figure supplement 5*) in both flies and mosquitoes. Similarly, *Cg25C*, *HPX4/pxn*, and *SCRBQ2/crq* are well-defined markers for granulocyte/plasmatocyte populations across species (*Figure 3—figure supplement 5*). However, while *PPO1* and *lozenge* are enriched in oenocytoid/crystal cell populations in our study and in *Drosophila* (*Tattikota et al., 2020*), these transcripts were respectively either in low abundance or were not detected in (*Figure 3—figure supplement 5*; *Raddi et al., 2020*).

Additional mosquito-specific immune cell markers identified in our study corresponding to prohemocyte (*ND2*) or granulocyte (*LRIM16A*, *SCRASP1*, *LRIM15*, and *LRR8*) populations displayed strong similarities to (*Figure 3—figure supplement 6*; *Raddi et al., 2020*). This is further supported by correlations of our prohemocyte and granulocyte cell clusters to comparable hemocyte subtypes defined in previous mosquito single-cell studies (*Figure 3—figure supplement 7*; *Raddi et al., 2020*). However, there are important distinctions in the markers used to denote oenocytoids between studies, where *PPO4* and *PPO9* used to delineate oenocytoids in previous studies (*Raddi et al., 2020*) contrast our results where *PPO4* and *PPO9* are expressed in both oenocytoid and granulocyte populations (*Figure 2G*, *Figure 3—figure supplement 6*) and have previously been implicated in phagocytic granulocytes (*Kwon and Smith, 2019*; *Smith et al., 2016*). Furthermore, the expression of *PPO8*, *SCRB3*, and *SCRB9* in oenocytoids that feature prominently in our analysis herein, are either found in low abundance or were not detected by (*Figure 3—figure supplement 6*; *Raddi et al., 2020*). There is also little similarity when our oenocytoid clusters (Clusters 7 and 8) and *Drosophila* crystal cell populations (*Figure 3—figure supplement 5*; *Cattenoz et al., 2020*; *Cho et al., 2020*; *Tattikota et al., 2020*) are compared to previously defined 'oenocytoid' cells (*Raddi et al., 2020*), which according to our analysis and others (*Hu et al., 2021*) more closely resemble granulocyte populations (*Figure 3—figure supplement 7*).

The patterns of *LysI* (AGAP011119) and *FBN10* (AGAP011230) used to respectively define PPO6-low and PPO6high immune cell populations in previous mosquito scRNA-seq studies (*Severo et al., 2018*) also provide significant comparative insight into the immune cell populations defined in our

study. We demonstrate that *LysI* is predominantly expressed in the phagocytic granulocyte populations of Clusters 2–4, while *FBN10* can be found in both granulocyte and oenocytoid populations (*Figure 2F*, *Figure 3—figure supplement 8*). There is a significant correlation of *LysI* and *FBN10* with *PPO6* expression (*Figure 3—figure supplement 8*), mirroring the PPO6$^{low}$ and PPO6$^{high}$ phenotypes as previously described (*Severo et al., 2018*), yet our data argue that these cell markers do not accurately account for the added complexity of mosquito immune cell populations identified in our study.

Together, these analyses highlight the similarities and differences between previous studies of insect hemocytes. Based on the homology of our cell clusters to *Drosophila* hemocytes (*Figure 2* and *Figure 3—figure supplement 5*), previous proteomic analysis of phagocytic hemocytes (*Figure 2—figure supplement 1*; *Smith et al., 2016*), and the functional assays that serve as confirmation of our cell types (*Figure 3*), we believe that these comparative analysis strengthens and further validates the identification of our cell clusters as prohemocytes, granulocytes, or oenocytoids.

## Differentiation of mosquito immune cell populations using lineage analysis

Previous studies in mosquitoes have suggested that prohemocyte precursors give rise to differentiated granulocyte and oenocytoid populations (*Ramirez et al., 2014*; *Rodrigues et al., 2010*; *Smith et al., 2015*). Additional evidence supports that granulocytes undergo mitosis to proliferate in response to infection (*King and Hillyer, 2013*). However, these observations have been based on morphological characterization, providing only speculation to the source of these immune cell populations. To better understand the origins of our identified immune cell clusters, we performed lineage analysis to determine relationships between the transcriptional profiles of individual cells to construct cell lineages in pseudotime using Monocle3 (*Cao et al., 2019*; *Packer et al., 2019*; *Trapnell et al., 2014*). Pseudotime analysis from naïve (*Figure 4A*), blood-fed (*Figure 4B*), or combined (naive and blood-fed) cell samples (*Figure 4C*) each reveal two distinct lineages from a shared precursor population (*Figure 4*). When visualized by cell cluster, these patterns support that the presumed prohemocyte precursors of Cluster 5 serve as the initial branching point for our cell lineages delineate into either granulocyte (Clusters 2, 3, 4, 6) or oenocytoid lineages (Clusters 7, 8; *Figure 4*). For the granulocyte lineage, precursor cells progress into immature granulocytes of Cluster 6 before maturation into the more specialized granulocyte populations of Cluster 2–4 (*Figure 4*). Pseudotime analysis suggests that Cluster 3 may represent an additional intermediate or transient cell-state in naive mosquitoes (*Figure 4A*), that is absent in cell populations from blood-fed conditions or may alternatively reflect changes in cytoadherence with different physiological conditions (*Figure 4B*, *Figure 1—figure supplement 3*). However, at present, we are unable to provide further resolution into the differentiation of these granulocyte populations without further detailed experiments.

For the oenocytoid lineage, Monocle3 analysis supports that precursor cells (Cluster 5) first differentiate into an intermediate, immature oenocytoid stage (Cluster 7) before maturation into the mature oenocytoid cells of Cluster 8 (*Figure 4*). Based on these cell trajectories (*Figure 4A–C*), as well as the transcriptional differences that likely define immature cell types (*Figure 3—figure supplement 4*), our results support a model for immune cell differentiation and the progression of cells within each lineage (*Figure 4D*). These data corroborate the differentiation of cells from a prohemocyte precursor as previously proposed (*Ramirez et al., 2014*; *Rodrigues et al., 2010*; *Smith et al., 2015*), while providing insight into the potential role of cell intermediates undergoing maturation before terminal differentiation of mosquito immune cell subtypes (*Figure 4D*) similar to those recently described in *Drosophila* (*Tattikota et al., 2020*).

## Lozenge promotes differentiation of the oenocytoid lineage

While several genes have been described that promote *Drosophila* immune cell lineages (*Evans et al., 2014*), the factors that define mosquito immune cell lineages have not been described beyond the role of multiple immune signaling pathways that influence hemocyte differentiation in response to malaria parasite infection (*Ramirez et al., 2014*; *Smith et al., 2015*). To further explore the factors that determine mosquito immune cell lineages, we focused on oenocytoid differentiation and the role of lozenge. In *Drosophila*, *lozenge* (lz) expression is integral to defining crystal cell fate (*Fossett et al., 2003*; *Waltzer et al., 2003*), the equivalent of mosquito oenocytoids. To similarly

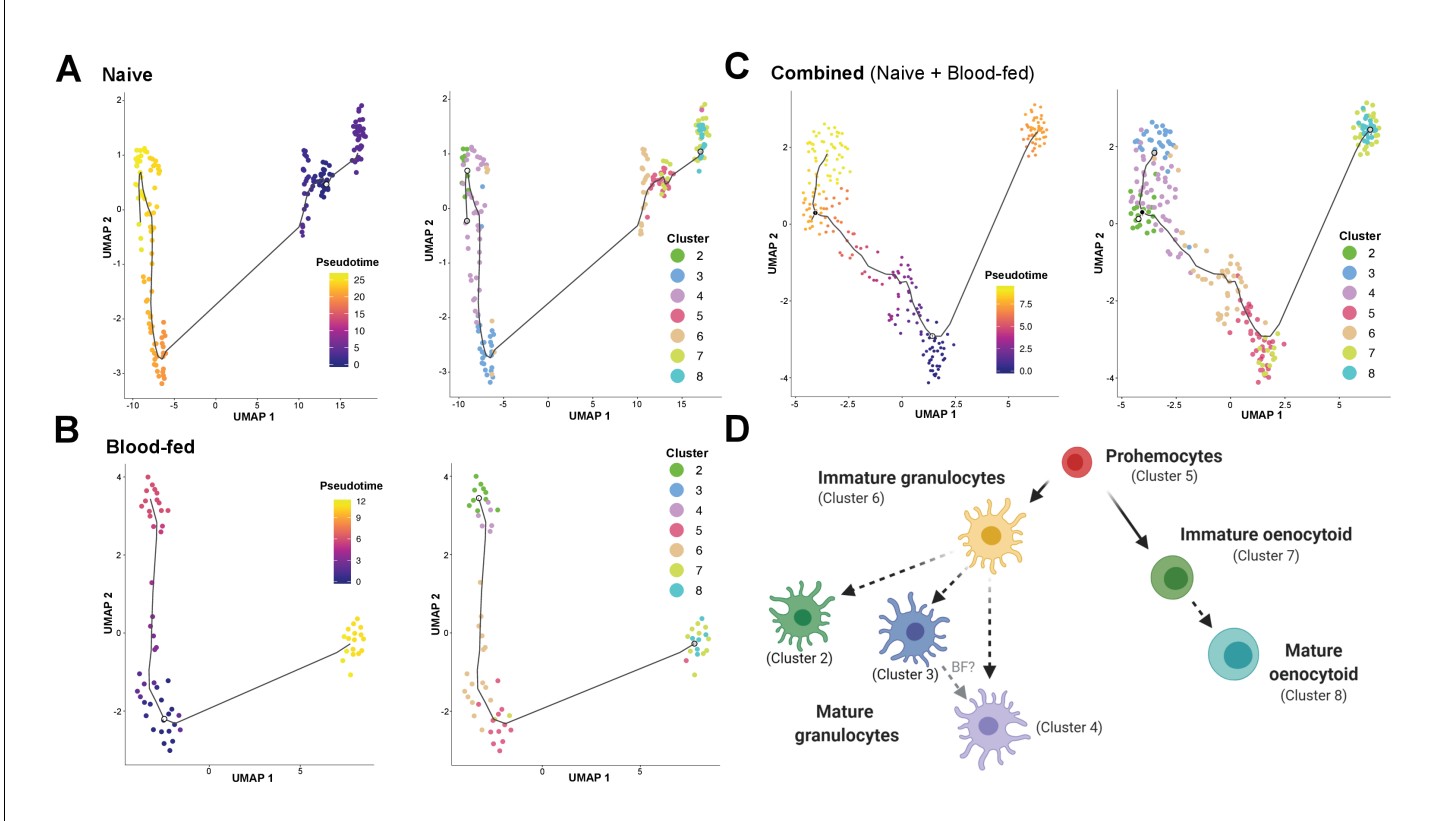

**Figure 4.** Lineage analysis of mosquito immune cells. Using Monocle3, mosquito immune cells were visualized by UMAP to reveal two distinct lineages in pseudotime under naive (A), blood-fed (B), or combined (naïve and blood-fed) samples (C) with the corresponding immune cell clusters for each condition. Based on the lineage analysis, gene expression, and other functional assays, our data support the following model of immune cell development and differentiation where prohemocytes serve as precursors for the granulocyte and oenocytoid lineages (D). Each cell type is labeled with the corresponding cell cluster described in our analysis. Figure was created with BioRender.com.

examine the role of lozenge in mosquito oenocytoid development, we used RNA-FISH to demonstrate and confirm the expression of *lozenge* in mosquito immune cells (*Figure 5A*). *Lozenge* was detected in ~15% of fixed cells (*Figure 5B*), a much higher percentage than that of $SCRB9^+$ cells demarcating *An. gambiae* oenocytoids (*Figure 3*). When we more closely examined the expression of *lozenge* and *SCRB9*, we see that co-localization of both markers only occurs in a subset of $lozenge^+$ cells (*Figure 5C*), suggesting that *lozenge* is expressed in other immune cell subtypes in addition to mosquito oenocytoid populations. This is supported by the expression of *lozenge* in other immune cell clusters (*Figure 5A*), the ability of a subset of lozenge+ cells to undergo phagocytosis, as well as the depletion of $lozenge^+$ cells and lozenge expression following depletion of phagocytic cell populations (*Figure 5—figure supplement 1*). To evaluate the influence of *lozenge* on oenocytoid cell fate, we silenced *lozenge* expression by RNAi (*Figure 5—figure supplement 2*) and examined the co-localization of *LRIM15/SCRB9* by RNA-FISH. In *lozenge*-silenced mosquitoes, we see a significant decrease in $LRIM15^-/SCRB9^+$ cells (*Figure 5D*), suggesting that *lozenge* is integral to defining the oenocytoid lineage. This is further supported by the specific reduction of *PPO1/3/8* expression (*Figure 5E*), PPOs that are enriched in Clusters 7 and 8 corresponding to the oenocytoid cell fate (*Figure 2F*). Together, these data support that *lozenge* expression is an important driver of the oenocytoid lineage in *An. gambiae* (*Figure 5F*). Based on our cell trajectories proposed in *Figure 4*, it suggests $lozenge^+$ prohemocytes promote the differentiation into an oenocytoid. However, the presence of *lozenge* in a subset of phagocytic granulocyte populations (*Figure 5*, *Figure 5—figure supplement 1*) may alternatively support a model of transdifferentiation in which oenocytoids can be derived from phagocytic granulocytes as previously proposed in *Drosophila* (*Leitão and Sucena, 2015*).

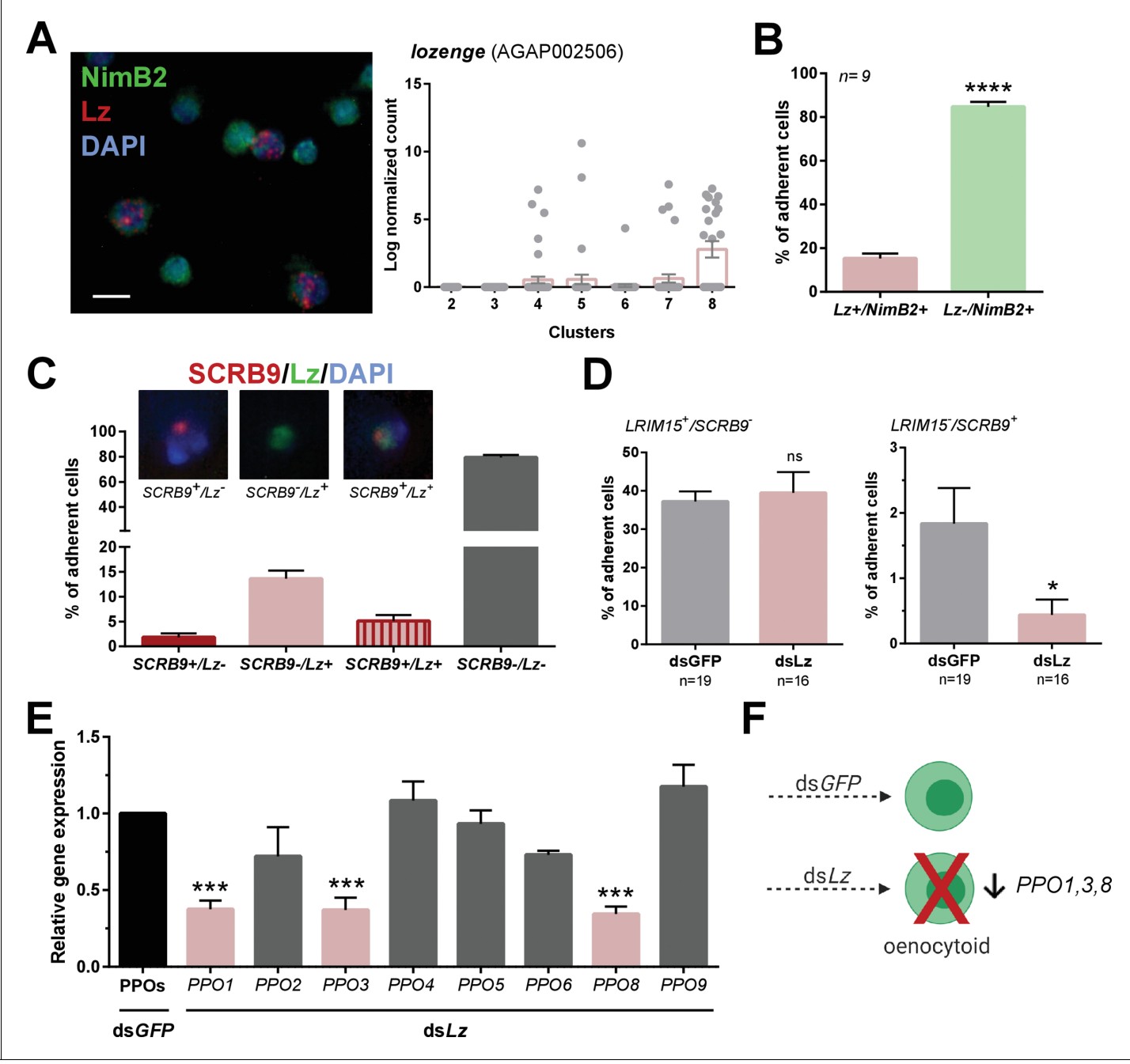

**Figure 5.** Lozenge promotes oenocytoid differentiation. RNA-FISH and gene expression profiles across cell clusters for *lozenge (Lz)* (A). Scale bar, 10 µm. The percentage of adherent *Lz+/NimB2+* or *Lz-/NimB2+* cells were examined in naïve adult female mosquitoes to estimate cell abundance (B). Data were collected from two independent experiments. Asterisks denote significance (****p< 0.0001). To more closely examine the population of *Lz+* cells, RNA-FISH experiments were performed double staining for *Lz* and the oenocytoid marker, *SCRB9* (C). The percentage of fixed cells positive for one, both, or neither marker is displayed with representative images. Data are summarized from two independent experiments. To determine the effects of Lz on immune cell populations, the abundance of *LRIM15+/SCRB9-* (granulocyte) and *LRIM15-/SCRB9+* (oenocytoid) cells were evaluated by RNA-FISH after *GFP* (control)- or *Lz*-silencing (D). Data represent the mean ± SE of three independent replicates. Significance was determined using Mann-Whitney analysis and is denoted by an asterisk (*p < 0.05); ns, not significant. Since Lz expression has previously been associated with prophenoloxidase (PPO) expression, the expression of all eight genes identified in our scRNA-seq analysis were examined by qRT-PCR in GFP (control) - *Lz*-silenced mosquitoes (E). Data represent the mean ± SE of three or more independent replicates and were analyzed by a one-way ANOVA and Holm-Sidak's multiple comparison test using GraphPad Prism 6.0. Asterisks denote significance (***p < 0.001). (F) Summary of *Lz*-silencing experiments which display a reduction in oenocytoid numbers and a specific sub-set of PPO gene expression which support that Lz is integral to the differentiation of the mosquito oenocytoid lineage.

*Figure 5 continued on next page*

*Figure 5 continued*

The online version of this article includes the following source data and figure supplement(s) for figure 5:

**Source data 1.** Source data pertaining to data presented in *Figure 5B–E*.
**Figure supplement 1.** Phagocytic properties of a subset of *lozenge (lz)*⁺ cells.
**Figure supplement 2.** Validation of *lozenge* knockdown following RNAi.

## Discussion

Our understanding of mosquito immune cells has largely been shaped by studies in other insects (*Banerjee et al., 2019*; *Lavine and Strand, 2002*). From morphological observations of size and shape, three cell types have been described in mosquitoes: prohemocytes, oenocytoids, and granulocytes (*Castillo et al., 2006*). However, with the advent of additional molecular tools to study mosquito immune cell function, several studies have supported an increased complexity of hemocyte populations beyond these generalized cell subtype classifications (*Bryant and Michel, 2016*; *Kwon and Smith, 2019*; *Pondeville et al., 2020*; *Raddi et al., 2020*; *Severo et al., 2018*; *Smith et al., 2016*). Herein, we demonstrate through scRNA-seq experiments and additional molecular characterization that there are at least seven conservatively defined immune cell populations in *An. gambiae*.

Similar to previous characterizations (*Castillo et al., 2006*), we identify prohemocyte, oenocytoid, and granulocyte populations in our RNA-seq analysis. Based on the lineage analysis, it would appear as though circulating prohemocytes can serve as progenitor populations that give rise to either the granulocyte or oenocytoid trajectories as previously proposed (*Castillo et al., 2006*; *Ramirez et al., 2014*; *Rodrigues et al., 2010*; *Smith et al., 2015*). However, in both the oenocytoid and granulocyte classifications, we identify multiple, distinct immune cell populations defined by developmental progression, activation state, or specialized immune function similar to those recently described in *Drosophila* (*Cattenoz et al., 2020*; *Tattikota et al., 2020*). As a result, the roles of oenocytoids and granulocytes may extend well beyond the respective oversimplified roles in melanization and phagocytosis that they have previously been ascribed (*Hillyer and Strand, 2014*; *Lavine and Strand, 2002*). Importantly, our experiments now provide a reliable set of markers to accurately distinguish between mosquito immune cell populations using RNA-FISH and validate these identifications through co-localization experiments, phagocytosis experiments, and phagocyte depletion assays. Therefore, our experiments provide an important foundation and much-needed cell markers to reliably distinguish oenocytoid and granulocyte populations that will advance the study of mosquito immune cells.

Through our scRNA-seq analysis, we identify at least four granulocyte subtypes in *An. gambiae* based on gene expression, previous proteomics studies (*Smith et al., 2016*), and phagocytic properties. This expansion of the general 'granulocyte' classification is supported by previous morphological observations of granulocytes in mosquitoes (*Kwon and Smith, 2019*; *Pondeville et al., 2020*), and more recently by parallel scRNA-seq experiments in *Drosophila* (*Cattenoz et al., 2020*; *Tattikota et al., 2020*) and *An. gambiae* (*Raddi et al., 2020*). Of these four granulocyte subtypes, our data support that cells of Cluster six are intermediate or immature granulocyte forms that display distinct expression patterns from prohemocyte precursors, yet do not have the same properties of other granulocyte subtypes (Cluster 2–4). This is supported by our pseudotime lineage analysis, where these immature cells of Cluster 6 give rise to more specialized granulocyte populations, similar to that described for comparable immature plasmatocytes in *Drosophila* (*Tattikota et al., 2020*). This maturation in Clusters 2–4 includes the increased expression of the phagocytic cell markers *LRIM15* and *SCRASP1* (*Smith et al., 2016*) as well as Cyclin G2 as a marker of differentiated cells (*Horne et al., 1997*; *Martínez-Gac et al., 2004*) that result in more specialized granulocyte subtypes.

Of these 'mature' granulocytes, Cluster two displays increased immune properties comparable to other recently described granulocyte or plasmatocyte populations in *An. gambiae* (*Raddi et al., 2020*) and *Drosophila* (*Cattenoz et al., 2020*; *Tattikota et al., 2020*). Based on their increased expression of antimicrobial peptides (AMPs) and other immune components such as TEP1, these cell populations may have a primary role in the hemocyte-mediated immune responses that limit bacteria (*Hillyer et al., 2003a*; *Reynolds et al., 2020*) or the recognition and killing of malaria parasites

(*Castillo et al., 2017*; *Kwon and Smith, 2019*). In addition, the increased expression of mitotic markers in Cluster 2 cells suggests that these granulocyte populations may contribute to hemocyte replication as previously suggested (*King and Hillyer, 2013*). Granulocytes of Cluster 3 can be delineated by the expression of *LRIM6*, as well as *cathepsin L* and *cathepsin F*. In other invertebrate systems, cathepsin L has been implicated in hemocyte lysosomes, serving important roles in the degradation of phagocytosed materials by phagocytic immune cells (*Jiang et al., 2018*; *Tryselius and Hultmark, 1997*). Both cathepsin L and cathepsin F have been associated with antimicrobial activity (*Guo et al., 2018*; *Jiang et al., 2018*), which together infer that these cells likely play an important role in immunity and immune homeostasis. However, it is at present unclear why these cells are only detected under naïve conditions. Lineage analysis of naïve cell populations suggests that Cluster 3 cells may represent a 'transition state' intermediates that gives rise to other granulocyte populations following blood-feeding or infection. Alternatively, the loss of Cluster 3 upon blood-feeding may also indicate changes in cytoadherence within these populations. Cells within Cluster 4 display high levels of *FBN10*, which resemble previously described PPO6$^{high}$ phagocytic cell populations (*Kwon and Smith, 2019*; *Severo et al., 2018*). However, additional studies are required to more closely resolve the impacts that feeding status (naive, blood-fed, *Plasmodium* infection) may have on these granulocyte subtypes as transient cell states or as differentiated cell types.

In addition to identifying multiple granulocyte subtypes, we also define two populations of mosquito oenocytoids (clusters 7 and 8) that likely reflect immature and mature cell populations analogous to those recently described for *Drosophila* crystal cell populations (*Cho et al., 2020*; *Koranteng et al., 2020*; *Tattikota et al., 2020*). Mature oenocytoids (Cluster 8) are denoted in part by the increased expression of *lozenge*, *PPO1*, *pebbled*, *DNA J*, *MLF*, *Notch*, and *klumpfuss* as previously described (*Cho et al., 2020*; *Koranteng et al., 2020*; *Tattikota et al., 2020*), as well as the increase in *SCRB3* and *SCRB9* which serve as a new marker of mosquito oenocytoids in our analysis. The mosquito oenocytoid lineage is distinct from that of granulocytes, relying on the expression of *lozenge* to promote oenocytoid differentiation similar to *Drosophila* crystal cells (*Fossett et al., 2003*; *Waltzer et al., 2003*). In *Drosophila*, *lozenge* is a transcription factor that interacts with the GATA factor *serpent* to promote the crystal cell lineage from embryonic or larval lymph gland prohemocytes (*Fossett et al., 2003*; *Waltzer et al., 2003*), a process that is tightly regulated by *u-shaped* expression (*Fossett et al., 2003*). In our analysis, *u-shaped* is expressed in mature granulocytes (Cluster 2–4), implying that its expression can be a marker of differentiated granulocytes that are no longer able to adopt an oenocytoid cell fate. This is supported by our on lineage analysis, where differentiation likely occurs from circulating prohemocyte precursor populations (Cluster 5), yet we cannot rule out the potential that immature granulocyte populations are able to undergo transdifferentiation as previously described in *Drosophila* (*Leitão and Sucena, 2015*). Together with our lozenge gene-silencing data, this suggests that the regulation of oenocytoid differentiation may be highly conserved between *Drosophila* and *Anopheles*.

Our study also breaks down existing paradigms that insect oenocytoids/crystal cells are primarily associated with prophenoloxidase (PPO) production and melanization (*Hillyer and Strand, 2014*; *Lavine and Strand, 2002*; *Lu et al., 2014*). This largely stems from work in *Drosophila*, where two of the three PPOs (*PPO1* and *PPO2)* are expressed in crystal cells (*Dudzic et al., 2015*), and in *Bombyx mori* where PPOs are exclusively synthesized by oenocytoids (*Iwama and Ashida, 1986*). However, our scRNA-seq results suggest that both granulocytes and oenocytoids are involved in PPO production, and that distinct subsets of PPOs are differentially regulated in the granulocyte and oenocytoid lineages. Following *lozenge*-silencing, we see significant decreases in *PPO1*, *PPO3*, and *PPO8* expression, transcripts that were highly enriched in oenocytoids in our study, while the remaining PPO genes were unaffected. This is supported by similar RNAi experiments in *Aedes aegypti*, where lozenge and Toll activation influence orthologous PPO gene expression (*Zou et al., 2008*). Recent studies of prostaglandin signaling in *An. gambiae* have also implicated the regulation of *PPO1*, *PPO3*, and *PPO8* in oenocytoid populations (*Kwon et al., 2020*), providing further support that a subset of PPOs are specifically regulated in oenocytoids. In addition, several lines of evidence support that granulocyte populations also contribute to PPO expression in *An. gambiae*, including PPO6 transgene expression and staining in mosquito granulocytes (*Bryant and Michel, 2016*; *Bryant and Michel, 2014*; *Castillo et al., 2006*; *Kwon and Smith, 2019*; *Severo et al., 2018*), the effects of phagocyte depletion on PPO expression (*Kwon and Smith, 2019*), and the identification

of multiple PPOs in the mosquito phagocyte proteome (*Smith et al., 2016*). This is a departure from other insect systems and is most likely a reflection of the expansion of the PPO gene family in mosquito species, where a total of nine *An. gambiae* PPOs have been annotated with yet undescribed function. Together, these results suggest new possible roles for PPOs in mosquito immune cells and their respective roles in the innate immune response.

Initial comparisons to recently published *Drosophila* immune cell scRNA-seq experiments (*Cattenoz et al., 2020*; *Tattikota et al., 2020*) reveal both similarities and differences in immune cell populations between dipteran species. Similar to *Drosophila*, our data in mosquitoes support the developmental progression and specialization of immune cells from precursor populations, the presence of multiple phagocytic cell populations, and multiple shared markers that delineate the oenocytoid/crystal cell lineage (*Cattenoz et al., 2020*; *Tattikota et al., 2020*). However, several differences are also noted in mosquitoes, including the absence of well-characterized *Drosophila* immune cell markers such as hemolectin (*Goto et al., 2003*; *Pondeville et al., 2020*) and hemese (*Kurucz et al., 2003*), as well as the lack of lamellocyte cell populations. We also identify several mosquito immune cell markers (such as LRIM15 and SCRB9) and the expansion of PPO genes in mosquito hemocytes that are unique to mosquito immune cell populations. Other respective differences in the isolation of cells from larvae or adults in *Drosophila* and mosquitoes, may further explain other disparities in cell types or steady states of activation. Ultimately, these questions require a more in-depth comparison of immune cells between these two organisms in the future.

When placed in the context of previously published scRNA-seq studies in mosquitoes (*Raddi et al., 2020*; *Severo et al., 2018*), our results provide additional resolution and perspective to the burgeoning study of mosquito immune cells. We expand upon the initial classification of PPO6[high] and PPO6[low] immune cell subtypes by *Severo et al., 2018*, providing an increased number of described hemocyte subtypes. Furthermore, our experiments support that the previously defined PPO6[high] and PPO6[low] populations (*Severo et al., 2018*) likely represent granulocyte subtypes based on their phagocytic ability and similarity to granulocyte gene expression profiles identified in our analysis. Similar to *Raddi et al., 2020*, we define prohemocytes, multiple granulocyte populations (including an immune-enriched subtype), and oenocytoids. While we identify comparable prohemocyte and granulocyte populations in our study, our studies significantly differ in the description of oenocytoid cell types. Based on our analysis, the oenocytoids defined by *Raddi et al., 2020* more closely resemble granulocytes, which lack conserved *PPO1* and *lozenge* markers of oenocytoid/crystal cell populations (*Benoit et al., 2017*; *Cattenoz et al., 2020*; *Cho et al., 2020*; *Tattikota et al., 2020*), as well as *SCRB3/SCRB9* markers that feature prominently in our analysis. Moreover, the oenocytoids described by Raddi et al. display limited expression of *PPO8* and lack *PGE2R* (*Raddi et al., 2020*), which have integral functional roles in oenocytoid immune cell function (*Kwon et al., 2020*). In addition, we do not detect signatures of megacytes, rare immune cell populations denoted by *TM7318* and *LL3* expression (*Raddi et al., 2020*). However, megacytes are enriched in blood- or *Plasmodium*-infected samples from 2 to 7 days post-feeding (*Raddi et al., 2020*), of which one caveat of our analysis is that we only examine immune cell population in naïve or blood-fed samples 24 hr post-feeding. While our analysis of Cluster 2 in our study suggests that these cells may have some proliferative properties, we do not detect the previously described proliferating granulocyte populations defined by *cyclin B* and *aurora kinase* expression (*Raddi et al., 2020*). However, these cells are enriched at later time points following blood-feeding or infection (*Raddi et al., 2020*), timepoints which are not included in our own study, similar to megacytes as described above. It is also unclear how technical differences in the experimental approach between studies (10x Genomics vs. FACS isolation followed by Smart-seq2 in our study) may have influenced these differences.

In contrast to the random isolation and cell sequencing of 10x Genomics methods employed by *Raddi et al., 2020*, the FACS-based methodology used in our study has enabled a more focused enrichment of immune cell populations based on DNA content and lectin staining. This has likely contributed to the resolution of the resulting hemocyte subtypes in our analysis despite sequencing a fraction of cells (262 versus 5383) when compared to previous studies (*Raddi et al., 2020*). Due to the increased scope of the study by Raddi et al. that examine multiple physiological conditions (naïve, blood-fed, *P. berghei*-infected) and experimental timepoints (days 0, 1, 2, 3, and 7) only 1127 (1049 naive and 78 one-day post-blood fed) of the 5383 total cells examined pair with the physiological conditions in our analysis, and may account for some differences between studies including the

absence of comparable megacyte and dividing granulocyte populations (*Raddi et al., 2020*) in our study.

A primary goal of our study has also been to integrate our dataset with previous descriptions of *Anopheles* hemocytes, thereby enhancing its role as a community resource by placing our analysis in the larger context of previously published work. As a result, we incorporate several markers detailed in previous immunofluorescence assays (*Bryant and Michel, 2016*; *Bryant and Michel, 2014*; *Castillo et al., 2006*; *Pinto et al., 2009*), transcriptional studies (*Pinto et al., 2009*), and proteomic analysis (*Smith et al., 2016*) that have proven instrumental to the characterization of our immune cell clusters. Strengthened by homology to studies of *Drosophila* hemocytes (*Cattenoz et al., 2020*; *Cho et al., 2020*; *Tattikota et al., 2020*), we provide a reliable set of lineage-specific markers that accurately define the granulocyte and oenocytoid lineages. When paired with phagocytosis assays and methods of phagocyte depletion (*Kwon and Smith, 2019*), we provide an enhanced set of tools to define mosquito immune cell populations. Through these new resources, our RNA-FISH data support that mosquito oenocytoid populations do not undergo phagocytosis, contrasting previous reports of rare phagocytic events by oenocytoids when relying on cellular morphology alone (*Hillyer et al., 2003b*). Therefore, our results improve upon existing knowledge and offer further advances to increase the consistency and resolution in the study of mosquito immune cells.

In addition, our study also provides several new insights into mosquito immune cell biology that warrant further study. This includes the role of ploidy in insect immune cell populations, the expression of CLIP-domain serine proteases (CLIPs) and serine protease inhibitors (SRPNs) in distinct oenocytoid and granulocyte populations, the enrichment of chemosensory genes in oenocytoids, and the variable expression of tRNAs across immune cell clusters. Together, these are suggestive of distinct transcriptional repertoires within immune cell subtypes that may impart as of yet unknown biological functions specialized for each cell type. Our data also provide additional detail into the regulation of immune cell differentiation, demonstrating the integral role of lozenge in driving the oenocytoid cell lineage. Other transcription factors such as LL3 (*Raddi et al., 2020*; *Smith et al., 2015*) and STAT-A (*Smith et al., 2015*) have been implicated in the differentiation of the granulocyte and oenocytoid lineages in *Anopheles*, yet at present, we have very little understanding of the signals that promote mosquito hematopoiesis and hemocyte differentiation. It is also of interest to examine how immune cell populations differ between life stages (larvae and adult), to determine how immune cells are influenced by the microbiota, how different physiological conditions mediate sessile hemocyte populations, and how different pathogen signatures can influence immune cell development and differentiation. As a result, we believe that the candidate markers and cell lineage progressions proposed by our study provide an essential first step to approach this multitude of questions in *Anopheles* hemocyte biology through future work.

To address many of these experimental questions, there is an inherent need to develop additional genetic tools and resources for the study of mosquito immune populations. This includes the development of transgenic lines expressing subtype-specific markers and binary expression systems similar to those developed for *Drosophila* (*Evans et al., 2014*), for which the results of our single cell transcriptomes and functional analysis will serve as an important foundation for future genetic studies to examine mosquito immune cell function.

In summary, our characterization of mosquito hemocytes by scRNA-seq and accompanying functional validation experiments provide an important advancement in our understanding of *An. gambiae* immune cell populations. Through the molecular characterization of at least seven immune cell subtypes and the development of dependable molecular markers to distinguish between cell lineages, we have presented new molecular targets where genetic resources were previously lacking. Through functional data and efforts to incorporate existing knowledge of mosquito hemocytes, we believe that our data will serve as an important resource for the vector community, which togethers offer new insights into the complexity of mosquito immune cells and provide a strong foundation for comparative functional analyses of insect immune cells.

## Materials and methods

**Key resources table**

*Continued on next page*

*Continued*

| Reagent type (species) or resource | Designation | Source or reference | Identifiers | Additional information |
|---|---|---|---|---|
| Reagent type (species) or resource | Designation | Source or reference | Identifiers | Additional information |
| Strain, strain background (*An. gambiae*) | Keele | *Hurd et al., 2005*; *Ranford-Cartwright et al., 2016* | | NA |
| Biological sample (*An. gambiae*) | adult female hemolymph | NA | NA | Perfused hemocytes from naïve (sugar-fed) or blood-fed (24 hr post-feeding) mosquitoes |
| Sequence-based reagent | LRIM15 (qRT-PCR primers) | *Smith et al., 2016* | AGAP007045 | **F**:CGATCCTGATCCTGAACGTGGGCTTC **R**:GCAAGCAAGCCACTCACAAATCCTCG |
| Sequence-based reagent | LRIM16A (qRT-PCR primers) | *Smith et al., 2016* | AGAP028028 | **F**:ATCAGAGTGCAGCACAAGTTGAAGGT **R**:TCTCTGTTAGCATAGCGCCTTCGTTC |
| Sequence-based reagent | Lz (qRT-PCR primers) | This study | AGAP002506 | **F**:GCACCGTCAATCAGAACCAA **R**:TGCCACTGATCGAATGCTTG |
| Sequence-based reagent | NimB2 (qRT-PCR primers) | *Kwon and Smith, 2019* | AGAP029054 | **F**:CAATCTGCTCAAATGGCTGCTTCCACG **R**:GCTGCAAACATTCGGTCCAGTGCATTC |
| Sequence-based reagent | PPO1 (qRT-PCR primers) | *Kwon and Smith, 2019* | AGAP002825 | **F**:GACTCTACCCGGATCGGAAG **R**:ACTACCGTGATCGACTGGAC |
| Sequence-based reagent | PPO2 (qRT-PCR primers) | *Kwon and Smith, 2019* | AGAP006258 | **F**:TTGCGATGGTGACCGATTTC **R**:CGACGGTCCGGATACTTCTT |
| Sequence-based reagent | PPO3 (qRT-PCR primers) | *Kwon and Smith, 2019* | AGAP004975 | **F**:CTATTCGCCATGATCTCCAACTACG **R**:ATGACAGTGTTGGTGAAACGGATCT |
| Sequence-based reagent | PPO4 (qRT-PCR primers) | *Kwon and Smith, 2019* | AGAP004981 | **F**:GCTACATACACGATCCGGACAACTC **R**:CCACATCGTTAAATGCTAGCTCCTG |
| Sequence-based reagent | PPO5 (qRT-PCR primers) | *Kwon and Smith, 2019* | AGAP012616 | **F**:GTTCTCCTGTCGCTATCCGA **R**:CATTCGTCGCTTGAGCGTAT |
| Sequence-based reagent | PPO6 (qRT-PCR primers) | *Kwon and Smith, 2019* | AGAP004977 | **F**:GCAGCGGTCACAGATTGATT **R**:GCTCCGGTAGTGTTGTTCAC |
| Sequence-based reagent | PPO8 (qRT-PCR primers) | *Kwon and Smith, 2019* | AGAP004976 | **F**:CCTTTGGTAACGTGGAGCAG **R**:CTTCAAACCGCGAGACCATT |
| Sequence-based reagent | PPO9 (qRT-PCR primers) | *Kwon and Smith, 2019* | AGAP004978 | **F**:TGTATCCATCTCGGACGCAA **R**:AAGGTTGCCAACACGTTACC |
| Sequence-based reagent | rpS7 (qRT-PCR primers) | *Kwon and Smith, 2019* | AGAP010592 | **F**:ACCCCATCGAACACAAAGTTGACACT **R**:CTCCGATCTTTCACATTCCAGTAGCAC |
| Sequence-based reagent | SCRB3 (qRT-PCR primers) | This study | AGAP005725 | **F**:CATCGGGACAGCTACATCCT **R**:TTATTGCTGCTACCGTTGCC |
| Sequence-based reagent | SCRB9 (qRT-PCR primers) | This study | AGAP004846 | **F**:CGATATTCGGCGATGCAACT **R**:CACGCATGACACGATTCAGT |
| Sequence-based reagent | GFP (T7 RNAi primers) | *Kwon and Smith, 2019* | NA | **F**:TAATACGACTCACTATAGGGAGAAT GGTGAGCAAGGGCGAGGAGCTGT **R**:CACGCATGACACGATTCAGT |
| Sequence-based reagent | Lz (T7 RNAi primers) | This study | AGAP002506 | **F**:TAATACGACTCACTATAGGGC TGCAACCGTCCCAGAACAACGGC **R**:TAATACGACTCACTATAGGG ACAAACCGGAGATCGTTGAATTTGG |
| Sequence-based reagent | Nimrod B2 (RNA-FISH probe) | Advanced Cell Diagnostics | AGAP029054 | *Severo et al., 2018* |
| Sequence-based reagent | LRIM15 (RNA-FISH probe) | Advanced Cell Diagnostics | AGAP007045 | regions 2–874 of XM_308718.4 |
| Sequence-based reagent | Lz (RNA-FISH probe) | Advanced Cell Diagnostics | AGAP002506 | regions 168–1372 of XM_312433.5 |
| Sequence-based reagent | SCRB3 (RNA-FISH probe) | Advanced Cell Diagnostics | AGAP005725 | regions 337–1276 of XM_315741.5 |

*Continued on next page*

*Continued*

| Reagent type (species) or resource | Designation | Source or reference | Identifiers | Additional information |
|---|---|---|---|---|
| Sequence-based reagent | SCRB9(RNA-FISH probe) | Advanced Cell Diagnostics | AGAP004846 | regions 402–1306 of XM_001688510.1 |
| Commercial assay or kit | Standard macrophage depletion kit | Encapsula NanoSciences LLC | CLD-8901 | Control liposomes or clodronate liposomes were used in a 1:5 dilution in 1x PBS |
| Commercial assay or kit | DNA Clean and Concentration kit | Zymo Research | D4013 | |
| Commercial assay or kit | MEGAscript RNAi kit | Life Technologies | AM1626 | |
| Commercial assay or kit | RevertAid First Strand cDNA Synthesis kit | Life Technologies | K1622 | |
| Commercial assay or kit | RNAscope Multiplex Fluorescent Detection Reagents V2 | Advanced Cell Diagnostics | 323110 | |
| Software, algorithm | Seurat | *Butler et al., 2018* | | |
| Software, algorithm | Monocle3 | *Cao et al., 2019* | | |
| Software, algorithm | alona | *Franzén and Björkegren, 2020* | | https://alona.panglaodb.se/ https://github.com/ oscar-franzen/alona/ |
| Software, algorithm | Graph Pad Prism | Graph Pad Software, LLC | | |
| Other | FITC-conjugated Wheat Germ Agglutinin (WGA) | Sigma | L4985 | 1:5000 |
| Other | DRAQ5 | Thermo Fisher Scientific | 62251 | 1:1000 |
| Other | Live/Dead Fixable Dead Cell Stain | Thermo Fisher Scientific | L34965 | 1:1000 |
| Other | FluoSpheres Fluorescent Microspheres | Molecular Probes | F8821, F8823 | Red or Green fluorescent fluorospheres for phagocytosis assays |
| Other | Opal Fluorophore reagent | Akoya Biosciences | Opal520 (FP1487001KT), Opal570 (FP1488001KT) | 1:1000 |
| Other | ProLongDiamond Antifade Mountant with DAPI | Life Technologies | P36966 | |
| Other | PowerUp SYBR Green Master Mix | Applied Biosystems | A25742 | |
| Other | E-RNAi | | | http://www.dkfz.de/ signaling/e-rnai3/idseq.php |
| Other | DRSC RNA Seq Explorer | *Tattikota et al., 2020* | | https://www.flyrnai.org/ scRNA/blood/ |
| Other | | *Raddi et al., 2020* | | https://hemocytes. cellgeni.sanger.ac.uk/ |
| Other | | This study | | https://alona.panglaodb.se/ results.html?job=2c2r1NM5Zl2qc W44RSrjkHf3Oyv51y_5f09d74b770c9 |

## Mosquito rearing

Adult *An. gambiae* mosquitoes of the Keele strain (*Hurd et al., 2005*; *Ranford-Cartwright et al., 2016*) were reared at 27°C with 80% relative humidity and a 14/10 hr light/dark cycle. Larvae were reared on a diet of fish flakes (Tetramin, Tetra), while adult mosquitoes were maintained on 10% sucrose solution and commercial sheep blood for egg production.

## Isolation and sorting of mosquito immune cells for single-cell RNA sequencing

Hemolymph was perfused from female mosquitoes (n=40) under naïve (3- to 5 day old) or blood-fed (~24 hr post-feeding) conditions using an anticoagulant solution as previously described (*Kwon and Smith, 2019*; *Smith et al., 2016*). Perfused hemolymph was diluted with 1X PBS to a total volume of 1 mL, then centrifuged for 5 min at 2000×g to pellet cells. After the supernatant was discarded, cells were washed two times in 1X PBS with an additional centrifugation step of 5 min at 2000×g between washing steps. Cells were incubated with WGA (1:5000, Sigma), DRAQ5 (1:1000, Thermo Fisher Scientific) and Live/Dead Fixable Dead Cell Stain (1:1000 Thermo Fisher Scientific) for 90 min at room temperature. Following incubation, cells were washed twice in 1X PBS to remove excess stain with a centrifugation step of 5 min at 2000 ×g and run on a BD FACSCanto cytometer (BD Biosciences). Based on the previous flow cytometry data for establishment of threshold values for gating (*Kwon and Smith, 2019*), cells smaller or larger than single cells were excluded. Cell viability was determined by the intensity of the blue fluorescent signal from the Live/Dead Fixable Dead Cell Stain, where dead cells display higher fluorescent signal. Following gating for cell viability, cell populations were distinguished by WGA and DRAQ5 signals and with individual cells sorted into each well of a 384-well plate (twintec PCR plates, Eppendorf, Germany) containing 2.3 µl lysis buffer (*Picelli et al., 2013*).

cDNA libraries were generated using a slightly modified version of Smart-seq2 as previously described (*Picelli et al., 2013*), where 23 cycles for cDNA amplification were used. Single-cell libraries were sequenced on the HiSeq2500 platform (Illumina) using 56 base pair single-end sequencing. Library preparation and sequencing was performed at ESCG and NGI, SciLifeLab, Sweden.

## Computational analysis

Sequencing reads were mapped to the *Anopheles gambiae* AgamP4 reference genome (Ensembl release 40) and ERCC sequences with the HISAT2 aligner version 2.1.0 (*Kim et al., 2015*). Only alignments with mapping quality 60 were retained. Quantification of gene expression was performed on the gene level. Overlapping exons of the same gene were merged based on their annotation coordinates. Counting of alignments on genome annotations was performed with the program subread version 1.6.2 (*Liao et al., 2014*) with the '-s' flag set to 0. Quality control of the data was performed by examining the fraction of sequencing output from ERCC templates versus the genome. Moreover, we applied a threshold of a minimum of 10,000 uniquely mapped reads per cell, only considering reads mapped in exons, that is intronic and intergenic reads are not counted toward the 10,000-minimum threshold; cells with fewer reads were not included in downstream analyses. Raw read counts were normalized to RPKM to adjust for gene length (*Mortazavi et al., 2008*). RPKM values were transformed with the function log2(**x**+1), where **x** is a vector of expression values. The complete quality-filtered gene expression data (as RPKM values) for all 262 cells is found in *Supplementary file 1*. Statistical analyses and data visualization were performed with the R software (http://www.r-project.org) version 3.5.3. Hierarchical clustering with Euclidean distance was used to define cell clusters; the ward.D2 agglomeration method was used for linkage. The final clusters were defined using a combination of manual examination of the tree structure and the cutreeDynamic function of the R package dynamicTreeCut (with method set to hybrid and deepSplit set to 4) (*Langfelder et al., 2008*).

## Clustering and functional analysis of single-cell data

Raw reads count for single cells data were normalized and scaled using scale factor function (log10) for genes of interest. The percentage and the average expression of the selected genes were calculated in a scaled normalized expression along a continuous color scale. Seurat Dotplot version (3.1.5) was used for visualization in R (version 3.5.3) software. The color intensity is proportional to a scaled average gene expression level for the selected genes across all clusters and the size of the circle is correspondence to the percentage of cells within each cluster expressing a gene (*Stuart and Satija, 2019*).

Heatmaps were produced using the pheatmap package (version 1.0.12), where the average expression of the selected genes was calculated from the normalized scaled RPKM values from clustering data. The working data frame matrix was prepared using tidy-verse package version (1.3.0)

for heatmap construction on selected gene sets corresponding (hemocyte gene orthologs, immune genes, etc.) to visualize expression across clusters. The clustering distance applied to the heatmaps was based on spearman.complete.obs with the scale set to be the selected genes as a comparison for visualization. Color intensity corresponds to the normalized and raw scaled average gene expression encoded with gray, white, red and firebrick 3, with the latter indicating an increase in the average expression level of a given gene within a cell cluster.

Cell cycle gene analysis was performed using the Seurat function 'AddModuleScore' to calculate the average expression levels of transcripts annotated to be involved in the cell cycle (GO:0007049) 191 transcripts similar to *Raddi et al., 2020*. After filtering out cells expressing less than 1% of the transcripts, 172 remaining transcripts were used in a combined expression score to calculate the enrichment of cell cycle genes among clusters. Positive scores indicate higher expression of genes involved in cell cycle regulation suggestive of cell proliferation.

To perform gene ontology (GO) analysis on each of the defined cell clusters, transcripts expressed in >80% of each respective cell cluster (*Supplementary file 4*) were examined to characterize the molecular composition of each cell type. Gene IDs (AGAP accession numbers) were classified based on gene ontology as previously (*Mendes et al., 2011*; *Smith et al., 2016*) to identify the functional categories of proteins within each cell cluster and to enable comparisons between cell clusters.

Differential expression analysis was performed using linear models as implemented in the alona software (https://github.com/oscar-franzen/alona/) (*Franzén and Björkegren, 2020*). Genes with false discovery rate (FDR; Benjamini-Hochberg's procedure) <5% and the absolute value of the log fold change>1.5 were considered significantly differentially expressed. All data can be visualized using the alona server (*Franzén and Björkegren, 2020*) at the following project link: https://alona.panglaodb.se/results.html?job=2c2r1NM5Zl2qcW44RSrjkHf3Oyv51y_5f09d74b770c9.

## Comparative analysis to other single-cell studies

Enriched gene sets corresponding to unique immune cell or non-hemocyte populations defined in previous single-cell studies for *An. gambiae* (*Raddi et al., 2020*) or *Drosophila* (*Tattikota et al., 2020*) were compared to each of the individual cell clusters defined in our analysis. Comparisons were based on presence/absence to genes with an averaged gene expression of >1 FKPM (*Supplementary file 5*) in our analysis, with the percentage of genes within the enriched gene sets used as a readout for comparison.

Additional comparisons of candidate genes across single-cell studies was performed by visualizing individual gene expression across tSNE maps as described above for our study, and compared to gene expression profiles produced using existing online resources for previous single-cell studies in *An. gambiae* (*Raddi et al., 2020*; https://hemocytes.cellgeni.sanger.ac.uk/) and *Drosophila* (*Tattikota et al., 2020*; https://www.flyrnai.org/scRNA/blood/).

## Cell trajectory and pseudotime analysis

Cells were assigned to cell groups using Monocle3 with UMAP clustering (*Cao et al., 2019*; *Packer et al., 2019*; *Qiu et al., 2017*; *Trapnell et al., 2014*). Data was normalized to remove batch effects using PCA clustering to 100 dimensions (*Haghverdi et al., 2018*). Pseudotime was calculated in Monocle3, with colors representing pseudotime changes among the cell clusters (*Cao et al., 2019*; *Packer et al., 2019*; *Qiu et al., 2017*; *Trapnell et al., 2014*). Bioinformatic methods for Monocle analyses can be found at: https://github.com/ISUgenomics/SingleCellRNAseq_RyanSmith, (*Smith, 2021*, copy archived at swh:1:rev:4b4b48d062ce112b9f53b5bbf43502d6cfae91a0).

## RNA-FISH

In order to classify hemocyte populations by detecting specific RNA expression, we used RNAscope-Multiplex Fluorescent Reagent Kit v2 Assay (Advanced Cell Diagnostics), and in situ hybridization was performed using the manufacturer's instruction. Using anticoagulant solution, hemolymph was perfused from non-blood fed mosquitoes (3–5 days old) and placed on a superfrost microscopic slide (Fisher Scientific) to adhere at RT for 20 min. Cells were fixed with 4% paraformaldehyde for 15 min at 4°C, then washed three times with 1X PBS. Hydrogen peroxide was applied to the cells, and slides were incubated for 10 min at room temperature (RT). After washing three times in sterile

distilled water, cells were treated Protease IV and incubated for 30 min at RT. To delineate hemocyte populations, a Nimrod B2 (AGAP029054) RNA probe conjugated with C1 (*Severo et al., 2018*) was used as a universal maker and was mixed with a specific RNA probe conjugated with C2, either leucine rich-repeat immune protein 15 (LRIM15: AGAP007045; regions 2–874 of XM_308718.4), or lozenge (Lz: AGAP002506; regions 168–1372 of XM_312433.5). For the identification of oenocytoid populations, SCRB3 (AGAP005725; regions 337–1276 of XM_315741.5) or SCRB9 (AGAP004846; regions 402–1306 of XM_001688510.1) RNA probes conjugated with C1 were co-incubated with either LRIM15 or Lz probes. All RNAscope probes are commercially available through Advanced Cell Diagnostics. Fixed hemocyte slides were hybridized with the respective mixtures of RNA probes in a HybEZOven for 2 hr at 40℃. After washed two times with wash buffer for 2 min, hybridized probes were incubated with respective AMP reagents (AMP1 and AMP2) for 30 min at 40℃ and with AMP3 for 15 min at 40℃. Cells were washed two times with wash buffer between AMP incubations. Cells were incubated with RNAscopeMultiplex FLv2 HRP-C1 for 15 min at 40℃, labeled with selected Opal Fluorophore reagent (Akoya Bioscience) at dilution facto (1:1000) for 30 min at 40℃ and treated with RNAscopeMultiplex FLv2 HRP blocker for 15 min at 40℃. Cells were washed two times with wash buffer for 2 min between incubations. Following C1 labeling, cells were incubated with a specific RNAscopeMultiplex FLv2 HRP-C2 conjugated solution, desired Opal Fluorophore reagent (1:1000; Opal520 or Opal570) and HRP blocker. Slides were initially treated DAPI (Advanced Cell Diagnostics) for 30 s at RT and mounted with ProLongDiamond Antifade Mountant with DAPI (Life Technologies). Cells displaying a positive signal were quantified as the percentage of positive cells of the total number of cells examined. Counts were performed from >50 adherent cells per mosquito from randomly chosen fields using fluorescence microscopy (Nikon Eclipse 50i, Nikon).

## Phagocytosis assays

Phagocytosis assays were performed by injecting 69 nl of 2% green fluorescent FluoSpheres (vol/vol) in 1X PBS to naïve female mosquitoes (3- to 5-day old) using a Nanoject II injector (Drummond Scientific). After injection, mosquitoes were kept at 27℃ for 2 hr before hemolymph was perfused on a superfrost slide. To define phagocytic cell populations, phagocytosis assays were paired with RNA-FISH experiments as described above using *LRIM15*, *SCRB9*, and *Lz* RNA probes. The proportion of phagocytic cells was quantified as the number of cells that had phagocytosed one or more beads of the total number of cells examined that displayed a positive signal for each of the respective RNA-FISH probes. Counts were performed from >50 adherent cells per mosquito from randomly chosen fields using fluorescence microscopy (Nikon Eclipse 50i, Nikon).

## Phagocyte depletion using clodronate liposomes

To confirm the identification of phagocytic cells from our defined immune cell clusters, validation experiments were performed using clodronate liposomes (CLD) to deplete phagocytic hemocytes (*Kwon and Smith, 2019*). Na-ve female mosquitoes (3- to 5-day old) were injected with either 69 nl of control liposomes (LP) or CLD (Standard macrophage depletion kit, Encapsula NanoSciences LLC) at 1:5 dilution in 1X PBS. At 24 hr post-injection, hemolymph was perfused, and RNA-FISH was performed to differentiate affected cell populations using RNA probes for NimB2 (C1), LRIM15 (C2), and Lz (C2) as described above. Hemocytes displaying a positive signal were quantified from 50 or more cells per mosquito.

## qRT-PCR

Gene expression analysis using qRT-PCR was performed to validate the influence of phagocyte depletion on non-phagocytic and phagocytic cells. cDNA was prepared from the previous studies (*Kwon and Smith, 2019*) corresponding to naïve adult female *An. gambiae* treated with either control liposomes (LP) or clodronate liposomes (CLD) 24 hr post-treatment. Hemocyte cDNA was prepared as previously described (*Kwon and Smith, 2019*) to analyze relative gene expression of *lozenge* between LP and CLD treatments. Specific transcripts representative of non-phagocytic and phagocytic cell populations were examined by qRT-PCR using primers listed in *Supplementary file 6*.

## Gene silencing by RNAi

RNAi experiments were performed as previously described (*Kwon et al., 2017*; *Kwon and Smith, 2019*; *Reynolds et al., 2020*; *Smith et al., 2016*; *Smith et al., 2015*). T7 primers for lozenge (Lz; AGAP002506) were designed using the E-RNAi web application (http://www.dkfz.de/signaling/e-rnai3/idseq.php) and listed in *Supplementary file 7*. T7 templates for dsRNA synthesis were prepared from amplified cDNA from 4 day old whole naïve mosquitoes. PCR amplicons were purified using the DNA Clean and Concentration kit (Zymo Research), and dsRNAs were synthesized using the MEGAscript RNAi kit (Life Technologies). Subsequent dsRNA targeting GFP (control) or Lz was resuspended in nuclease-free water to 3 μg/μl after ethanol precipitation. Injections were performed in 3- to 4-day-old cold anesthetized mosquitoes by intrathoracic injection with 69 nl (~200 ng) of dsRNA per mosquito using a Nanoject III. The effects of gene silencing were measured at 3 days post-injection in whole mosquitoes (n=15) by qRT-PCR as previously described (*Kwon and Smith, 2019*).

## Acknowledgements

We thank Shawn Rigby of the Iowa State Flow Cytometry Facility for his assistance with FACS as well as Anna-Maria Divne at the Microbial Single Cell Genomics (MSCG) facility, SciLifeLab, for consultation regarding the sorting of individual hemocytes, the Eukaryotic Single Cell Genomics (ESCG) facility, SciLifeLab, for assistance with single-cell library preparation and The National Genomics Infrastructure, SciLifeLab, Sweden for assistance with Illumina sequencing. We also want to thank Maiara Severo-Witte and Elena Levashina for initial assistance with the RNA-FISH methods, Rick Masonbrink and Andrew Severin of the Iowa State Genome Informatics Facility for assistance with the single-cell RNA-seq analysis. Lastly, we would like to acknowledge the Swedish National Infrastructure for Computing (SNIC) for data handling and preprocessing of the scRNA-seq raw data for this study, which is partially funded by the Swedish Research council through grants agreement 2018–05973 to JA. This work was supported by the Swedish Society for Medical Research (SSMF) and the Swedish Research Council (VR-NT) to JA, the Agricultural Experiment Station at Iowa State University and the National Institutes of Health, National Institute of Allergy and Infectious Diseases (R21 AI44705) to RCS.

## Additional information

### Funding

| Funder | Grant reference number | Author |
|---|---|---|
| Swedish Society for Medical Research | | Johan Ankarklev |
| Swedish Research Council | | Johan Ankarklev |
| National Institute of Allergy and Infectious Diseases | R21AI144705 | Ryan C Smith |

The funders had no role in study design, data collection and interpretation, or the decision to submit the work for publication.

### Author contributions

Hyeogsun Kwon, Ryan C Smith, Conceptualization, Data curation, Formal analysis, Supervision, Funding acquisition, Validation, Investigation, Visualization, Methodology, Writing - original draft, Project administration, Writing - review and editing; Mubasher Mohammed, Data curation, Formal analysis, Investigation, Visualization, Methodology, Writing - review and editing; Oscar Franzén, Data curation, Software, Formal analysis, Validation, Investigation, Visualization, Methodology, Writing - review and editing; Johan Ankarklev, Conceptualization, Data curation, Software, Formal analysis, Supervision, Funding acquisition, Validation, Investigation, Visualization, Methodology, Project administration, Writing - review and editing

**Author ORCIDs**
Hyeogsun Kwon (iD) https://orcid.org/0000-0002-4141-4061
Oscar Franzén (iD) http://orcid.org/0000-0002-7573-0812
Ryan C Smith (iD) https://orcid.org/0000-0003-0245-2265

**Ethics**
Animal experimentation: The protocols and procedures used in this study were approved by the Animal Care and Use Committee at Iowa State University (IACUC-18-228).

**Decision letter and Author response**
Decision letter https://doi.org/10.7554/eLife.66192.sa1
Author response https://doi.org/10.7554/eLife.66192.sa2

## Additional files

**Supplementary files**

• Supplementary file 1. FKPM values of individual immune cells following scRNA-seq analysis.

• Supplementary file 2. Differential gene expression in immune cell clusters displaying significant differences between naive and blood-fed cells.

• Supplementary file 3. Significant markers of immune cell clusters identified by the FindAllMarkers program using the Seurat toolkit.

• Supplementary file 4. Genes expressed in more than >80% of cells within each respective immune cell cluster.

• Supplementary file 5. Averaged gene expression of cells within each immune cell cluster.

• Supplementary file 6. Primers for qRT-PCR and dsRNA-mediated gene silencing.

• Supplementary file 7. Primers for RNAi.

• Transparent reporting form

**Data availability**

Data generated and analysed in this study are included in the manuscript and supporting files. In addition, data can be visualized and downloaded using the following server: https://alona.panglaodb.se/results.html?job=2c2r1NM5Zl2qcW44RSrjkHf3Oyv51y_5f09d74b770c9.

The following dataset was generated:

| Author(s) | Year | Dataset title | Dataset URL | Database and Identifier |
|---|---|---|---|---|
| Smith RC | 2021 | JA_SCrna_mosq_hemocytes | https://alona.panglaodb.se/results.html?job=2c2r1NM5Zl2qcW44RSrjkHf3Oyv51y_5f09d74b770c9 | alona, 5dcbe6ad781464be604a43505a2fef18 |

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
