## [Decision Letter]

**Acceptance summary:**

In this manuscript, Kwon et al., describe a single cell sequencing analysis of hemocytes of the malaria mosquito Anopheles gambiae. Their data support the old classification of hemocytes in the three main categories previously defined according to the morphology and phenotype of these cells, but they further reveal subpopulations in the granulocyte and oenocytoid groups. The authors also provide several new markers to define these different populations and subpopulations.

**Decision letter after peer review:**

[Editors’ note: the authors submitted for reconsideration following the decision after peer review. What follows is the decision letter after the first round of review.]

Thank you for submitting your work entitled "Single-cell analysis of mosquito hemocytes identifies signatures of immune cell sub-types and cell differentiation" for consideration by *eLife*. Your article has been reviewed by 3 peer reviewers, and the evaluation has been overseen by a Reviewing Editor and a Senior Editor.

Our decision has been reached after consultation between the reviewers. Based on these discussions and the individual reviews below, we regret to inform you that your work will not be considered at this stage for publication in *eLife*. Nevertheless, the reviewers agree that they could consider a revised version of the manuscript that addresses all their points. So you have the opportunity to submit a revised version of this paper to *eLife* but this revised version will be considered as a new submission (but likely handle by the same reviewers). The editor and reviewers find many merits to your article but one first point is to know what it specifically brings compared to other RNA seq done with hemocytes of mosquitoes. A second point is that your study is rather descriptive. This not a problem by itself if you can provide a solid dataset that can serves of reference for the whole community (see criticisms of reviewer 2 on this point).

Essential revisions:

*Reviewer #1:*

Mosquito hemocyte biology and function remains poorly understood. However these cells have critical functions for example in immunity and there is a knowledge gap that needs filled. Several subtypes -three in total have been defined in the past but this study goes deeper into the functional characteriusa characterisation based on phagocytic properties, FISH and importantly single cell RNA sequencing. Moreover the importance of Lozenge in differentiation of oenocytoid hemocytes is shown. This is an elegant study relevant to the field. It will certainly impact studies on hemocytes on other mosquitoes and beyond. The introduction is clear and explains the necessity if this study. Methods are described well enough to follow what was done. Data can be accessed in a single cell analysis pipeline which is very useful to end users. I also like that blood fed and non blood fed mosquitoes were used, broadening impact and interest. Overall the manuscript does an excellent job of illustrating expression patters and differences between cells, including lineage analysis and conclude that 7 different types of immune cells, including 4 types of granulocytes. The definition of markers will make it easy for others interested in this topic to translate these findings into different questions, and signficantly expands our understanding of hemocytes in an important disease vector.

The following should be clarified or expanded on:

1) Are the three cell types 1 figure 1A meant to represent the three subtypes or these just three distinct gated populations as separated by FACS- I appreciate that the authors state that the subgroups/clusters realate to these populations but is there a selection into the three main known types? What are the cell types that were not split into these three groups- is there any indication of their identity?

2) Could the authors add information on lozange functions and how it may promoter oenocytoid differentiation? Even expanding on this in the Discussion would be useful for the reader who is not familiar with the topic.

3) Could the sequences of probes for RNA fish be indicated?

4) Line 554: could the authors expand on the question of "suitability" of methods used? This might save others in the field time when analysing similar data.

5) Line 531: "." instead of ".." after Sweden.

6) Line 922: An. Gambiae

7) Immune gene activation is seen in a subset of cells, could the authors speculate on where these may be part of an ongoing immune response to infection; and whether they would expect these genes to be expressed in mosquitoes kept in presence of antibiotics?

*Reviewer #2:*

Kwon et al. analyse the immune cell diversity of the adult mosquito Anopheles gambiae using single cell sequencing. They bled naïve and blood-fed adult females and sorted the hemocytes using WGA and DRAQ5 as hemocyte markers. Cells were selected from three independent gates defined by the intensity of DRAQ5, and sequenced using SMART-seq2 methodology. 262 cells were sequenced and included in the downstream analysis. The authors found 8 clusters of hemocytes using hierarchical clustering. Cluster 3 seems specific to naïve hemocytes, whereas all other clusters are found in both naïve and blood fed conditions. They dismissed the cluster 1 based on high gene expression, which may be indicative of FACS doublet. Next, the authors described the markers common to all hemocytes and the ones distinguishing the clusters. They identify clusters 2, 3 and 4 as granulocytes, 7 and 8 as oenocytoids and 5 as prohemocyte. Then, they use Monocle 3 to predict the filiation between the clusters based on a limited number of cluster markers.

This work represents a c database of hemocyte subtypes in mosquitoes and has the potential to highlight strong markers for each subpopulation. However, the experimental design is not explained properly and the authors mostly described their data without deep interpretation. Moreover, the definition of each subtype is not convincing. For these reasons, I believe that the work does not warrant publication in *eLife* in principle.

1) The number of sequenced cells seems rather low, which may bias the overall interpretation of the data.

2) The authors should provide the rational to look at naïve versus blood fed hemocytes. In addition, merging the two sets of data may again bias their interpretation. Typically, cluster 3 is only present in naïve hemocytes and cluster 2 is overrepresented in blood-fed hemocytes.

3) The biological meaning of the gating strategy for the FACS is not interpreted nor justified. What is the meaning of Clusters 3, 4 and 6 being excluded from gate 1?

4) What is DRAQ5? There is no description nor reference to the use of DRAQ5 to label hemocytes.

5) The justification for removing Cluster 1 from the analysis is not convincing: "we dismissed this cell cluster from further analysis as likely cell doublets of mixed cell origins as the result of FACS cell isolation or as dividing cells". If these cells were indeed doublets of mixed cells, they should not cluster together and if they were dividing cells, markers of division should be present. The authors should provide more evidence to remove cluster 1 or include it in the downstream analysis. As a matter of fact, the transcriptional profile suggests that these cells may represent pluripotent precursors for granulocytes and oenocytoid cells.

6) Cluster 5 is defined as prohemocyte based on the absence of Cyclin G2. Most hemocyte markers are also absent from this cluster (Figure 2C,D). What are the evidence that this cluster is indeed populated by hemocytes? Did the authors search for mitotic markers?

7) Most markers presented across the different figures are expressed in several clusters. The author should provide a figure or table displaying the markers expressed in a single cluster. Such table can be generated using the Seurat toolkit with the FindAllMarkers program (https://satijalab.org/seurat/). Figure 2A shows the level of expression of different genes and the percentage of cells expressing them. This is a more correct representation of the sc RNA data compared to that in the following panels (2 C-E), where there is no information as to the number of cells within the cluster expressing a given gene. Based on Figure 2A, the identification of different clusters does not seem to rely on robust criteria. For example, Cluster 2 to 4 seem very similar. Altogether, very few markers are taken into consideration.

8) Lines 190-200: the authors state that "PPO1, PPO3, and PPO8 are enriched in putative oenocytoids, while PPO2, PPO4, PPO5, PPO6, and PPO9 are most abundant in putative granulocyte populations (Figure 2E)", in contrast to previous suggestions that PPOs are expressed in a subset of hemocytes. This sentence is not in agreement with the data, Figure 2E highlights three main clusters expressing distinct pattern of PPOs, suggesting that indeed, distinct subsets of hemocytes express specific PPOs. But again, this panel does not show the percentage of cells expressing the different genes. We only get this information for PPO1, which is by the way only expressed in 25% of the cluster 8 cells.

9) The data shown in several supplementary figures do not seem to add much information, as such.

10) What is the evidence for cluster 4 producing cluster 2 and 3 (which is also specifically present in one condition)?

11) Figure 2A,C and 5A: lozenge ID is changing from 002506 in A to 002825 in C. In addition, the expression profiles are not concordant between the two graphs. In A, we observe a strong expression in clusters 5, 7 and 8, while in C we only see expression in cluster 8. Additionally, the expression of Lz shown in Figure 5A (histogram on log normalized count) suggest more cells strongly expressing Lz in cluster 8 and 4 compared to cluster 5, which is again different from the observation in Figure 2A.

12) Monocle 3 analysis. On which basis was done the selection of the genes for the Monocle 3 analysis (Figure S14)? The Dot plot indicates that most genes are strongly expressed across several clusters. How is it possible to infer filiation based on ubiquitously expressed genes? In addition, the number of genes seems extremely low. Can the analysis be done on the whole expression matrix?

13) Monocle 3 interpretation. In Figure 4B, cluster 7 is split into two cell groups, one joined with the prohemocyte cluster 5 and the other with cluster 8. Thus, the group joined with cluster 5 could constitute the progenitors of cluster 8 independently from cluster 5. What are the evidence of the link between cluster 5 and cluster 7?

*Reviewer #3:*

In this manuscript, Kwon et al. describe a single cell sequencing analysis of hemocytes of the malaria mosquito Anopheles gambiae. Their data support the old classification of hemocytes in the three main categories previously defined according to the morphology and phenotype of these cells, but they further reveal subpopulations in the granulocyte and oenocyte groups. The authors also provide several new markers to define these different populations and subpopulations. Of note, two additional scRNAseq studies on A. gambiae/coluzzii hemocytes are available: Severo et al., PNAS 2018, and Raddi et al., BioRxiv 2020. Still, while overlapping to some extend (especially Kwon and Raddi), I believe that the three papers reinforce each other, each of them bringing a different perspective.

Compared to Severo 2018, Kwon et al. selected a larger diversity of hemocytes as they did not restrict themselves to PPO6 expressing hemocytes, and thus, they were able to get more diverse transcriptomic clusters covering most hemocyte types. Of note, they sequenced ~10x more cells than Severo et al., however the coverage was much lower with a smaller reads (56 vs 100 bases) and a relatively low cutoff for minimal read number (10 000 reads per cell while Severo et al. had several millions reads per cell). Still, this lower coverage did not affect their cell classification, but likely restricted it to highly expressed genes.

Raddi et al. sequenced an even larger number of cells (~20x more compared to Kwon et al), which allowed them to identify some additional subpopulations, and especially one, the megacytes that was not at all described in Kwon et al. While Raddi et al. focused on hemocyte changes after Plasmodium infection, Kwon et al. characterised the phagocytic properties of their different subpopulations.

[Editors’ note: further revisions were suggested prior to acceptance, as described below.]

Thank you for submitting your article "Single-cell analysis of mosquito hemocytes identifies signatures of immune cell sub-types and cell differentiation" for consideration by *eLife*. Your article has been reviewed by 2 peer reviewers, and the evaluation has been overseen by a Reviewing Editor and Utpal Banerjee as the Senior Editor. The following individuals involved in review of your submission have agreed to reveal their identity: Angela Giangrande (Reviewer #1); Stéphanie Blandin (Reviewer #2).

The reviewers have discussed the reviews with one another and the Reviewing Editor has drafted this decision to help you prepare a revised submission. The reviewers have the feeling that you can addressed the revisions without further experiments. Thus the revisions requested below only address clarity and presentation.

Summary:

In this manuscript, Kwon et al., describe a single cell sequencing analysis of hemocytes of the malaria mosquito Anopheles gambiae. Their data support the old classification of hemocytes in the three main categories previously defined according to the morphology and phenotype of these cells, but they further reveal subpopulations in the granulocyte and oenocytoid groups. The authors also provide several new markers to define these different populations and subpopulations.

Essential revisions:

1) The reasons given for removing cluster 1 from subsequent analyses are not acceptable.

The authors bring the following arguments: (i) "Cluster 1 does not express specific marker and express markers also present in other clusters." From the presented figure, it appears that Cluster 1 expresses at least three specific markers (CDC42, 005742 and 004634 in Figure 2A) that are completely absent from the other clusters. (ii) "Cluster 1 is an outlier in the t-SNE analysis." The distance between dots in the t-SNE analysis depends on the projection that was done and is merely indicative of the distance between the clusters. The hierarchical clustering provided by the authors in Figure 1B is more quantitative in terms of homology between clusters and clearly shows the proximity of Cluster 1 with Clusters 2 to 4. Moreover, the distance between Cluster 1 and 2-4 is shorter than the distance between 7 and 8 that are both oenocytoids. (iii) "Cluster 1 presents a high median of gene number." Figure 1D shows that the number of genes is in the same range than Cluster 8. (iv) Based on the above statements, the authors believe that Cluster 1 represents cell doublets. However, an efficient gating strategy on the FACS should remove doublets. In addition, we would expect at least a higher DRAQ5 signal for doublets (higher DNA content), but Cluster 1 is also found among cells gated with gate 2 and gate 3. At last, if these cells were doublets, their specific markers should be present in other clusters as well (e.g. CDC42, 005742 and 004634). (v) The authors exclude the possibility that these cells are transdifferentiating by the lack of mitotic markers. However, transdifferentiation does not necessarily involve cell division and the two processes are independent in *Drosophila* for both crystal cells and lamellocytes. (vi) The authors exclude the possibility that these cells represent megacytes based on the expression of two markers. Raddi et al., (2020) provide a list of 102 markers for megacytes. The authors should at least provide the data for all the markers (in the form of a Dotplot for example).

Cluster 1 should be included in all graphical representation of Figure 2 as well as in the lineage analysis of Figure 4. Browsing through the web application of the authors, one can notice notably that Cluster 1 is enriched for LRIM15 and thus could be considered as another cluster of phagocytes?

2) The definition of Cluster 5 as prohemocytes still relies exclusively on the absence of markers (Figure S10).

Raddi et al., (2020) defined two populations of prohemocytes. The authors should at least show the comparison of the markers for cluster and the prohemocytes markers indicated in Raddi et al., report. Also, the presence of Clusters 5 and 7 could reflect the presence of two prohemocyte populations, resembling the situation observed in *Drosophila* (see the work of L Waltzer). At this, point, the interpretation of the data should be more cautious.

3) Blood-fed vs. naïve conditions.

It would help to analyze the data in either naïve or in blood-fed conditions, rather than merging the data from the two conditions. As shown in Figure S3, the eight clusters are represented in very different manner in the two conditions, from 0% Cluster 3 to almost 50% Cluster 2 in blood-fed animals.

The interpretation of the impact of blood feeding on the hemocytes (cluster 2 and 4) should be described in the result section, possibly with a figure displaying the differentially expressed genes (present in Table S2).

Of note, only Cluster 2 hemocytes are enriched in cell cycle gene: how many genes, in how many cells of the cluster? Can they be preferentially ascribed to a food regimen? Is it possible to compare Cluster 2 to the proliferative cluster observed in Raddi et al.?

4) A resource manuscript should provide useful information to the community. This manuscript would gain from a more systematic comparison with the data already available in the literature. For example, Raddi et al., define PPO 4 and 9 as being characteristic of oenocytoids.

5) Could the authors explain why there are no cells from the blood-fed condition in Gate 4?

6) On DRAQ5 labelling used for the FACS sorting step, in Figure 1A, several levels of DRAQ5 are observed in the dot plots. Other publications mention the use of DRAQ5 to estimate the ploidy of the cells. Is this what we observe on the dotplots in Figure 1A, the first three gates separating cells based their ploidy? Is this known for mosquitoes' hemocytes? How do the authors interpret this, since cluster 8 is exclusively found in gate 1?

7) line 1048: "higher levels".

8) Figure 2E: "Ninjurin" is Ninjirin in the text.

9) Figure S14: Are the data normalised by column or are these expression levels? The unit of the colour gradient should be mentioned (z-score, expression levels?).

10) I would appreciate a somehow more detailed comparison of their hemocyte categories (including cluster 1) with those from Raddi et al., This could be proposed as a supplementary figure for instance. Is there a 1 to 1 correlation between the categories? Also, please indicate discrepancies when relevant, e.g. PPO4 is used as a marker for oenocytoids in Raddi et al. while this gene seems to be expressed in all hemocyte subgroups, and especially in granulocytes, in Kwon et al.

11) The discussion paragraph where the authors compare their work with the two other scRNAseq reports (l519-544) is somehow awkward. While I do understand the need to provide an overview of the three studies, I would rather insist there (1) on the reason why they managed to recover as many clusters as Raddi et al., despite sequencing fewer cells (strategies not compared in the text), (2) on a more precise comparison of the hemocyte clusters from the 3 studies (see previous point), and (3) summarising current knowledge on the functional characterisation of the different hemocyte categories.

12) Changes in cluster populations upon blood feeding (e.g. disappearance of all hemocytes from cluster 3): this could be due to rewiring/changes in gene expression in some specific cells as suggested by the authors. Another explanation could be a change in adherence: if certain cells become sessile after blood feeding, they will not be recovered during mosquito perfusion. Of note, it is unclear why there are no hemocytes identified in gate 4 after blood feeding. As this gating is not selective, one would have expected to recover cells there in both sugar fed and blood fed conditions.

13) Figure S1: the cluster color code is different from that of Figure 1. Please use the same for all figures.

---

## [Author Response]

[Editors’ note: the authors resubmitted a revised version of the paper for consideration. What follows is the authors’ response to the first round of review.]

Essential revisions:Reviewer #1:Mosquito hemocyte biology and function remains poorly understood. However these cells have critical functions for example in immunity and there is a knowledge gap that needs filled. Several subtypes -three in total have been defined in the past but this study goes deeper into the functional characteriusa characterisation based on phagocytic properties, FISH and importantly single cell RNA sequencing. Moreover the importance of Lozenge in differentiation of oenocytoid hemocytes is shown. This is an elegant study relevant to the field. It will certainly impact studies on hemocytes on other mosquitoes and beyond. The introduction is clear and explains the necessity if this study. Methods are described well enough to follow what was done. Data can be accessed in a single cell analysis pipeline which is very useful to end users. I also like that blood fed and non blood fed mosquitoes were used, broadening impact and interest. Overall the manuscript does an excellent job of illustrating expression patters and differences between cells, including lineage analysis and conclude that 7 different types of immune cells, including 4 types of granulocytes. The definition of markers will make it easy for others interested in this topic to translate these findings into different questions, and signficantly expands our understanding of hemocytes in an important disease vector.The following should be clarified or expanded on:1) Are the three cell types 1 figure 1A meant to represent the three subtypes or these just three distinct gated populations as separated by FACS- I appreciate that the authors state that the subgroups/clusters realate to these populations but is there a selection into the three main known types? What are the cell types that were not split into these three groups- is there any indication of their identity?

We would like to thank the reviewer for this comment. In our experience using flow cytometry on mosquito hemocyte populations, we have consistently noticed distinct “groups” of hemocytes based on their cellular properties defined by the intensity of wheat germ agglutinin (WGA, a general marker of hemocytes) and DNA signal (DRAQ5). Since WGA and DRAQ5 are both non-specific markers, we have been unable to further define these hemocyte populations without additional markers that could delineate prohemocyte, granulocyte, and oenocytoid cell types.

As a result, we decided to use these previous flow cytometry observations to add an additional component of “selection/enrichment” in our cell isolations using FACS. Therefore, we selected for “gates” to target each of the three major groups (Gates 1-3) identified from flow cytometry, as well as an additional non-selective (Gate 4) gate (outlined in Figure 1A and Figure S2). Although we did not know which hemocyte sub-types were reflected in our arbitrary FACS gating’s, we thought that the scRNA-seq information gathered from these populations could be informative. As shown in Figure S2, Gate 1 predominantly enriched our oenocytoid populations (Clusters 7 and 8), while Gate 3 enriched for our mature granulocyte populations (Cluster 2-4). Cells of Clusters 5 and 6, defined as prohemocytes and immature granulocytes in our study, were found most abundant in Gate 2 and the non-selected Gate 4 (Figure 2). Of note, cells collected with Gate 4 (non-selective) captured cells from each of our hemocyte clusters.

Therefore, the result of our FACS gating experiments demonstrate the power and promise to define mosquito immune cell populations in the absence of cell-type specific markers. This strategy has enabled us to sample and profile the extent of mosquito immune cells, enriching for cells based on cell properties rather than by random abundance as in Drop-seq methods. This has greatly facilitated our study to accurately profile individual immune cell subtypes with a lower number of individual cells. We believe that this ability to crudely distinguish cell types by FC can similarly inform studies in other species in the absence of cell markers.

As the reviewer has noted, we do apply some minimal bias in our selection, in which immune cells were selected for by their WGA^+^ signal, a general marker indicative of mosquito hemocytes (Castillo et al. 2006). As a result, we did not examine the identity of the WGA^-^ cells since they likely represent the presence of bacteria or other contaminants commonly associated with hemolymph perfusion techniques. We also do not find evidence of the megacyte defined by Raddi et al. 2020. At present it is unclear if our FACS strategy may have selected against this rare cell type.

We have amended our manuscript to incorporate many of the ideas described above in the Results section to better address our methodology (Lines 85-93 and 111-118).

2) Could the authors add information on lozange functions and how it may promoter oenocytoid differentiation? Even expanding on this in the Discussion would be useful for the reader who is not familiar with the topic.

We would like to thank the reviewer for the excellent suggestion, and apologize for the omission of a discussion of lozenge in the discussion of our original manuscript.

Outside of studies in *Drosophila*, very little has been done to examine lozenge function in other insects. Only a single paper has implicated lozenge in the regulation of PPO gene expression and influence on malaria parasite infection in mosquitoes (Zou et al. 2008), therefore making our findings with lozenge the first to implicate its role in mosquito oenocytoid differentiation. While further work is needed to examine lozenge function in mosquitoes, our findings are heavily supported by previous studies in *Drosophila* demonstrating an integral role of lozenge in promoting crystal cell differentiation (Fossett et al. 2003, Waltzer et al. 2003), the equivalent of oenocytoids in mosquitoes.

In our revised manuscript, we have added a paragraph placing our findings regarding lozenge in the larger context of studies that have delineated its important role in hemocyte differentiation. This can be found on Lines 455-460.

3) Could the sequences of probes for RNA fish be indicated?

We would like to thank the reviewer for this suggestion. Due to the propriety nature of the RNA-FISH probes provided by Advanced Cell Diagnostics the exact sequences of the probes are unknown, however in the revised manuscript we have included the regions for the probe design from our manuscript in Lines 690-695. In addition, the probes used in our analysis are commercially available.

4) Line 554: could the authors expand on the question of "suitability" of methods used? This might save others in the field time when analysing similar data.

It is unclear what was originally intended in this description of the methodology used in our study. We have removed this sentence from the revised manuscript.

5) Line 531: "." instead of ".." after Sweden.

Thank you for catching this oversight. This has been corrected in the revised manuscript.

6) Line 922: An. Gambiae

Thank you for catching this oversight. This has been corrected in the revised manuscript.

7) Immune gene activation is seen in a subset of cells, could the authors speculate on where these may be part of an ongoing immune response to infection; and whether they would expect these genes to be expressed in mosquitoes kept in presence of antibiotics?

Thank you for the suggestion. We believe that the dataset generated by our scRNA-seq study provides a strong foundation for a wealth of questions regarding mosquito immune cell function. This includes addressing the very questions that the reviewer mentions regarding the immune response to infection and the influence of the microbiota on immune cell development, of which we are very interested in pursuing in future experiments.

At present, the model is that a subset of immune cells are “activated” to promote basal levels of immune expression and serve as immune sentinels for pathogen defense. There is evidence that mosquito immune cell activation is transient (Bryant et al., 2014, 2016) and responds to physiological changes that coincide with blood-feeding (Reynolds et al. 2020). However, it is unclear how the mosquito microbiota influence hemocyte expression. We hope to explore these questions through experiments using a combination of antibiotics (as suggested) and/or the production of axenic mosquitoes that have undergone development in the absence of bacteria. While the potential outcomes are unclear for mosquitoes, the presence of symbiotic bacteria in Tsetse flies has proven essential for immune cell development (Benoit et al. 2017).

As requested by the reviewer, we have amended the discussion in our revised manuscript to add speculation regarding these topics of interest (lines 560-566).

Reviewer #2:Kwon et al. analyse the immune cell diversity of the adult mosquito Anopheles gambiae using single cell sequencing. They bled naïve and blood-fed adult females and sorted the hemocytes using WGA and DRAQ5 as hemocyte markers. Cells were selected from three independent gates defined by the intensity of DRAQ5, and sequenced using SMART-seq2 methodology. 262 cells were sequenced and included in the downstream analysis. The authors found 8 clusters of hemocytes using hierarchical clustering. Cluster 3 seems specific to naïve hemocytes, whereas all other clusters are found in both naïve and blood fed conditions. They dismissed the cluster 1 based on high gene expression, which may be indicative of FACS doublet. Next, the authors described the markers common to all hemocytes and the ones distinguishing the clusters. They identify clusters 2, 3 and 4 as granulocytes, 7 and 8 as oenocytoids and 5 as prohemocyte. Then, they use Monocle 3 to predict the filiation between the clusters based on a limited number of cluster markers.This work represents a c database of hemocyte subtypes in mosquitoes and has the potential to highlight strong markers for each subpopulation. However, the experimental design is not explained properly and the authors mostly described their data without deep interpretation. Moreover, the definition of each subtype is not convincing. For these reasons, I believe that the work does not warrant publication in eLife in principle.

We would like to thank the reviewer for their thoughtful review. We have made significant efforts to address the comments and suggestions that were raised in the original review of our manuscript, and believe that the changes in our revised manuscript should now satisfy your criticisms regarding our experimental design, methods, and data visualization.

1) The number of sequenced cells seems rather low, which may bias the overall interpretation of the data.

We respectfully disagree with the reviewer’s assessment. While we understand that the 262 cells examined in our study are not an exhaustive analysis of mosquito immune cell populations, this is a significant improvement over the 24 cells previously examined by Severo et al. (2018) that was published after the onset of our own experiments.

While it is true that the number of cells examined is an important variable for any single cell study to achieve proper depth, there are also large differences in the number of cells examined depending on methodology. Drop-seq methodologies are less sensitive in their analysis, resulting in the identification of fewer genes detected per cell (Ziegenhain et al. 2017), which requires larger cell numbers to increase coverage. This contrasts the Smart-seq2 methods used in our analysis, which comparative analysis supports that Smart-seq2 is the most sensitive methodology for scRNA-seq, resulting in the detection of the highest number of genes per cell and the most even coverage across transcripts (Ziegenhain et al. 2017). As a result, far fewer cells are required using Smart-seq2 methods when compared to Drop-seq to achieve comparable resolution and insight into single-cell populations.

Moreover, our FACS gating strategy enabled the enrichment of more rare cell types based on characteristics for lectin staining (WGA) and DNA content (DRAQ5) that may have otherwise been masked by more abundant cell types using drop-seq methods. Therefore, we believe that our resulting analysis is a robust profile of mosquito immune cell sub-types.

This is evident when comparing our study to a recently published study by Raddi et al. (2020) that similarly examines mosquito immune cells. Using Drop-seq methods, they define ~5300 immune cells from multiple time points and physiological conditions. While our study lacks similar breadth, our study does provide comparable resolution into immune cell sub-types in *An. gambiae*, that when paired with the additional functional characterization in our study, we believe provides a superior community resource.

2) The authors should provide the rational to look at naïve versus blood fed hemocytes. In addition, merging the two sets of data may again bias their interpretation. Typically, cluster 3 is only present in naïve hemocytes and cluster 2 is overrepresented in blood-fed hemocytes.

We would like to thank the reviewer for their comment. Since blood-feeding represents a major physiological event for mosquitoes and with existing evidence supporting that blood-feeing promotes changes to mosquito hemocyte populations (Bryant et al., 2014, 2016; Castillo et al. 2011; Reynolds et al. 2020), we felt it was important to assess hemocytes under both naïve and blood-fed conditions to perform comparative analyses of cell activation and cell fate activated due to blood-feeding. We have added text to the Results section in our revised manuscript to provide additional rational for our experimental methodology (lines 82-83).

In regard to our analysis of the merged naive and blood-fed data, we respectfully disagree that it may bias the interpretation of our data. Admittedly, the mosquito hemocyte field is still in its infancy, where the lack of genetic tools/resources and cell markers have limited our understanding of mosquito immune cells. As a result, the premise of our scRNA study is more discovery based, which has the promise to serve as the foundation for future studies. For this reason, we believe it is important to examine all cells together, independent of their physiological context. As the reviewer mentions, we believe it is of interest that cells of Cluster 3 are only present under naïve conditions. However, at present, we are unable to determine if this is a transient cell state or a terminally differentiated cell type. We hope to further investigate this possibility, but believe that is well beyond the scope of this study.

Cluster 2 is found in both naïve and blood-fed conditions but does not seem to be significantly overrepresented in blood-fed conditions. Future work is required to more closely quantify how these “immune-activated” cell types respond to different physiological conditions and infection states.

We have also amended our discussion to enhance our discussion of these two cell clusters in the context of mosquito physiology (lines 420-443).

3) The biological meaning of the gating strategy for the FACS is not interpreted nor justified. What is the meaning of Clusters 3, 4 and 6 being excluded from gate 1?

We would like to thank the reviewer for their comment. Similar to the comments raised by R1, it has become apparent that we did not adequately explain the rationale behind our gating strategy used in our experiments and apologize for the omission.

Our gating strategy was based on published (Kwon and Smith, 2019) and unpublished observations using flow cytometry with a general lectin stain (WGA) and DNA stain (DRAQ5), in which we have consistently observed sub-populations with different cell properties. As a result, we selected for cells that correspond to each of the observed sub-types (ultimately gates 1-3) and used these for our gating experiments. Our hope was that through the scRNA-seq analysis we could define the cells responsible for these observable flow cytometry phenotypes.

Therefore, in response to the reviewer’s question, we did not knowingly exclude cells belonging to any of the 8 identified cell clusters in our analysis. In fact, the selection of cells in our gating strategy likely enhanced our ability to detect less abundant cells. This is evident by Gate 1, which represent less abundant cell types (based on intensity of counts by flow cytometry), and enriches for Clusters 1, 7, and 8. These properties are primarily based on DRAQ5 signal, which serves as a measurement of DNA content in our analysis. We have amended the Results section in our revised manuscript (Lines 85-93 and 111-118) to better convey these findings.

4) What is DRAQ5? There is no description nor reference to the use of DRAQ5 to label hemocytes.

DRAQ5 is a commonly used flow cytometry/FACS far-red stain to measure DNA content. This cell-permeable dye can be used to stain live or fixed cells and is not specific for mosquito hemocyte populations. We have added text to make this more transparent in our revised manuscript (Lines 86-87).

5) The justification for removing Cluster 1 from the analysis is not convincing: "we dismissed this cell cluster from further analysis as likely cell doublets of mixed cell origins as the result of FACS cell isolation or as dividing cells". If these cells were indeed doublets of mixed cells, they should not cluster together and if they were dividing cells, markers of division should be present. The authors should provide more evidence to remove cluster 1 or include it in the downstream analysis. As a matter of fact, the transcriptional profile suggests that these cells may represent pluripotent precursors for granulocytes and oenocytoid cells.

We would like to thank the reviewer for their comment and have taken steps in our revised manuscript to hopefully better convince the reviewer and future readers of these aberrant cell populations. As demonstrated in the revised Figure 2A, the inclusion of “cluster-specific” markers identified with the Seurat FindAllMarkers program argue that Cluster 1 strongly expresses all genes examined at relatively high levels. This includes otherwise strong identifiers of granulocyte or oenocytoid cell lineages. In our revised manuscript we provide additional text to support that these could be cells undergoing trans-differentiation of a granulocyte into an oenocytoid as previously described in *Drosophila* (Leitao et al. 2015), yet also provide support that the lack of increased cell division markers makes this far less likely. Other scRNA-seq studies of insect immune cells have yet to identify a pluripotent precursor population, nor do they seem to be the recently discovered populations of megacytes (Raddi et al. 2020), such that the more likely and more conservative approach is to assume that these are doublets containing both granulocyte and oenocytoid cell types. In the revised manuscript we have included additional text to better support our justification to exclude this cell population from further analysis (lines 129-147).

6) Cluster 5 is defined as prohemocyte based on the absence of Cyclin G2. Most hemocyte markers are also absent from this cluster (Figure 2C,D). What are the evidence that this cluster is indeed populated by hemocytes? Did the authors search for mitotic markers?

In our revised manuscript, we have taken additional steps to better convey to the reader the set of markers that define the prohemocytes of Cluster 5, as well as other immune cell clusters, in our analysis. We believe that the changes to our revised Figure 2 and the inclusion of a new supplemental figure (Figure S10) better highlight candidate marker genes for each of our defined cell populations.

While the reviewer is correct in that cells of Cluster 5 are absent for many of our described hemocyte markers, they are WGA+ (a general lectin stain indicative of mosquito hemocytes; Kwon and Smith, 2019) used in our FACs isolation and also express universal hemocyte markers such as NimB2, PPO6, SPARC, and Cg25C. The absence of well-defined granulocyte and oenocytoid markers as well as cyclin G2 expression is indicative that these cell population are likely undifferentiated cells. A cursory analysis of genes associated with the cell cycle in *Anopheles* (GO:0007049) as performed in Raddi et al. (2020) do not display significant differences across our immune cell clusters.

7) Most markers presented across the different figures are expressed in several clusters. The author should provide a figure or table displaying the markers expressed in a single cluster. Such table can be generated using the Seurat toolkit with the FindAllMarkers program (https://satijalab.org/seurat/). Figure 2A shows the level of expression of different genes and the percentage of cells expressing them. This is a more correct representation of the sc RNA data compared to that in the following panels (2 C-E), where there is no information as to the number of cells within the cluster expressing a given gene. Based on Figure 2A, the identification of different clusters does not seem to rely on robust criteria. For example, Cluster 2 to 4 seem very similar. Altogether, very few markers are taken into consideration.

We would like to thank the reviewer for the suggestion. In our revised manuscript, we have used the FindAllMarkers program in Seurat to identify a set of markers that more accurately define each our cell clusters in Figure 2A. This encompasses a total of 26 genes, with between 2-4 “markers” per cluster. As a result, we believe that the revised Figure 2A better encapsulates the similarities and differences between our defined immune cell clusters. We have also incorporated similar “bubble” plots as in Figure 2A for the remaining panels of Figure 2 in our revised manuscript.

8) Lines 190-200: the authors state that "PPO1, PPO3, and PPO8 are enriched in putative oenocytoids, while PPO2, PPO4, PPO5, PPO6, and PPO9 are most abundant in putative granulocyte populations (Figure 2E)", in contrast to previous suggestions that PPOs are expressed in a subset of hemocytes. This sentence is not in agreement with the data, Figure 2E highlights three main clusters expressing distinct pattern of PPOs, suggesting that indeed, distinct subsets of hemocytes express specific PPOs. But again, this panel does not show the percentage of cells expressing the different genes. We only get this information for PPO1, which is by the way only expressed in 25% of the cluster 8 cells.

We would like to thank the reviewer for this comment. After reviewing this section of text, it is obvious that we did not clearly articulate our intended message. Based predominantly from work in *Drosophila*, the mosquito community has long adopted the belief that PPOs were only produced in oenocytoids, the equivalent of *Drosophila* crystal cells. We have amended this section of text to better reflect our original intentions that mosquito PPOs are expressed in each of the major immune cell sub- types, with specific PPOs expressed in each of the respective prohemocyte, granulocyte, and oenocytoid cell types (Lines 205-219). We have also addressed the reviewers comment regarding the display of the gene expression data by modifying Figure 2F (and the rest of Figure 2) to reflect gene expression and the percentage of cells expressing each respective PPO as a “bubble” graph.

9) The data shown in several supplementary figures do not seem to add much information, as such.

While we appreciate the critique, without additional information regarding the specific figures in question it is difficult to make changes to our revised manuscript. We do however make changes to the supplemental information to more streamline the presentation of data. In some cases, this means the addition of new figures to better represent candidate marker genes identified in our analysis, as well as the consolidation/deletion of previous supplemental figures. While some of the current files may be descriptive or simplistic in the data that they convey, we believe that they are essential to our characterization of mosquito immune cells. In addition, we have added text to the discussion of our revised manuscript to better integrate these descriptive results into the experimental outcomes for these mosquito immune cell populations (lines 548-554).

10) What is the evidence for cluster 4 producing cluster 2 and 3 (which is also specifically present in one condition)?

We would like to thank the reviewer for the comment. At present, we have little biological evidence aside from the Monocle 3 lineage analysis predictions to prove that cells defined in C4 appear to be an intermediate state to those represented in C2 and C3, or perhaps cells transitioning between C2 and C3. While we have amended Figure 4 and the discussion of our revised manuscript (lines 335-337), we believe that further validation of these immune cell validations is beyond the initial characterizations of these immune cell clusters by scRNA-seq provided in our analysis. With potential years of future study to precisely delineate these questions, we have chosen our language very carefully to avoid absolute terms in describing these predicted analyses.

11) Figure 2A,C and 5A: lozenge ID is changing from 002506 in A to 002825 in C. In addition, the expression profiles are not concordant between the two graphs. In A, we observe a strong expression in clusters 5, 7 and 8, while in C we only see expression in cluster 8. Additionally, the expression of Lz shown in Figure 5A (histogram on log normalized count) suggest more cells strongly expressing Lz in cluster 8 and 4 compared to cluster 5, which is again different from the observation in Figure 2A.

We would like to apologize for this labeling error for the lozenge gene accession number. This has been corrected in the revised manuscript. Moreover, these discrepancies can be explained by differences in the display of the data, moving between bubble graphs and heat maps in our analysis. However, with modifications to Figure 2 in our revised manuscript, we believe that we have addressed these differences in display to offer a more congruent visualization of data across figures.

12) Monocle 3 analysis. On which basis was done the selection of the genes for the Monocle 3 analysis (Figure S14)? The Dot plot indicates that most genes are strongly expressed across several clusters. How is it possible to infer filiation based on ubiquitously expressed genes? In addition, the number of genes seems extremely low. Can the analysis be done on the whole expression matrix?

We would like to thank the reviewer for the comment. Our Monocle 3 analysis was performed as suggested on the whole expression matrix and have removed the data initially shown in Figure S14 in our original submission to remove any confusion in our revised manuscript.

13) Monocle 3 interpretation. In Figure 4B, cluster 7 is split into two cell groups, one joined with the prohemocyte cluster 5 and the other with cluster 8. Thus, the group joined with cluster 5 could constitute the progenitors of cluster 8 independently from cluster 5. What are the evidence of the link between cluster 5 and cluster 7?

We would like to thank the reviewer for their comments and for their observation. Based on our clustering analysis, t-SNE results, and candidate marker expression indicating that these clusters comprise independent cell populations, we believe that this split of Cluster 7 cells in the Monocle 3 pseudotime projections supports that these cells have recently begun the process of oenocytoid differentiation from prohemocyte precursors (Cluster 5). Our expression analysis suggest that Cluster 7 represents immature oenocytoids that are intermediates to the fully mature oenocytoids of Cluster 8, following similar immune progressions to orthologous crystal cells in *Drosophila* (Tattikota et al. 2020). We do provide experimental data that lozenge-silencing reduces the percentage of oenocytoids, but these data only indirectly demonstrate this association. With the markers identified in our study, we hope to directly approach this question in the future to gain a more definitive understanding of hematopoiesis and immune cell differentiation in *Anopheles*. However, with potentially years of work required to fully address this question, we believe that it is beyond the scope of our current study.

Reviewer #3:In this manuscript, Kwon et al. describe a single cell sequencing analysis of hemocytes of the malaria mosquito Anopheles gambiae. Their data support the old classification of hemocytes in the three main categories previously defined according to the morphology and phenotype of these cells, but they further reveal subpopulations in the granulocyte and oenocyte groups. The authors also provide several new markers to define these different populations and subpopulations. Of note, two additional scRNAseq studies on A. gambiae/coluzzii hemocytes are available: Severo et al., PNAS 2018, and Raddi et al., BioRxiv 2020. Still, while overlapping to some extend (especially Kwon and Raddi), I believe that the three papers reinforce each other, each of them bringing a different perspective.Compared to Severo 2018, Kwon et al. selected a larger diversity of hemocytes as they did not restrict themselves to PPO6 expressing hemocytes, and thus, they were able to get more diverse transcriptomic clusters covering most hemocyte types. Of note, they sequenced ~10x more cells than Severo et al., however the coverage was much lower with a smaller reads (56 vs 100 bases) and a relatively low cutoff for minimal read number (10 000 reads per cell while Severo et al. had several millions reads per cell). Still, this lower coverage did not affect their cell classification, but likely restricted it to highly expressed genes.Raddi et al. sequenced an even larger number of cells (~20x more compared to Kwon et al), which allowed them to identify some additional subpopulations, and especially one, the megacytes that was not at all described in Kwon et al. While Raddi et al. focused on hemocyte changes after Plasmodium infection, Kwon et al. characterised the phagocytic properties of their different subpopulations.

We would like to thank the reviewer for the kind words. We strongly agree with the reviewer’s assessment that our study complements those of other studies, each building off of each other and providing a unique perspective to enhance our understanding of mosquito hemocytes. We also believe that the functional validation of our cell types and descriptive nature of our study provide a more valuable and accessible resource than the previous studies.

[Editors’ note: what follows is the authors’ response to the second round of review.]

Essential revisions:1) The reasons given for removing cluster 1 from subsequent analyses are not acceptable.The authors bring the following arguments: (i) "Cluster 1 does not express specific marker and express markers also present in other clusters." From the presented figure, it appears that Cluster 1 expresses at least three specific markers (CDC42, 005742 and 004634 in Figure 2A) that are completely absent from the other clusters. (ii) "Cluster 1 is an outlier in the t-SNE analysis." The distance between dots in the t-SNE analysis depends on the projection that was done and is merely indicative of the distance between the clusters. The hierarchical clustering provided by the authors in Figure 1B is more quantitative in terms of homology between clusters and clearly shows the proximity of Cluster 1 with Clusters 2 to 4. Moreover, the distance between Cluster 1 and 2-4 is shorter than the distance between 7 and 8 that are both oenocytoids. (iii) "Cluster 1 presents a high median of gene number." Figure 1D shows that the number of genes is in the same range than Cluster 8. (iv) Based on the above statements, the authors believe that Cluster 1 represents cell doublets. However, an efficient gating strategy on the FACS should remove doublets. In addition, we would expect at least a higher DRAQ5 signal for doublets (higher DNA content), but Cluster 1 is also found among cells gated with gate 2 and gate 3. At last, if these cells were doublets, their specific markers should be present in other clusters as well (e.g. CDC42, 005742 and 004634). (v) The authors exclude the possibility that these cells are transdifferentiating by the lack of mitotic markers. However, transdifferentiation does not necessarily involve cell division and the two processes are independent in *Drosophila* for both crystal cells and lamellocytes. (vi) The authors exclude the possibility that these cells represent megacytes based on the expression of two markers. Raddi et al., (2020) provide a list of 102 markers for megacytes. The authors should at least provide the data for all the markers (in the form of a Dotplot for example).Cluster 1 should be included in all graphical representation of Figure 2 as well as in the lineage analysis of Figure 4. Browsing through the web application of the authors, one can notice notably that Cluster 1 is enriched for LRIM15 and thus could be considered as another cluster of phagocytes?

We would like to thank the reviewers for their comments. This section of text has been deleted in our revised manuscript and have taken a more “data-driven” approach to examine Cluster 1. In our revised manuscript, we now include additional analysis that suggest that cells in Cluster 1 are contaminants (fat body, etc.) that are common in perfused hemolymph samples (Castillo et al. 2006, Smith et al., 2016). Similar non-hemocyte cell types have been identified in other hemocyte single cell studies (Raddi et al., 2020, Tattikota et al., 2020), and have performed comparative analysis that argue that our Cluster 1 cells most likely represent these non-hemocyte contaminants. These comparative analyses have been included in Figure 2—figure supplement 2 and have been addressed in lines 177 to 197 of our revised manuscript.

Due to the high probability that these cells are non-hemocyte in origin, we do not believe that Cluster 1 should be included in the further representation of hemocyte markers in Figure 2 or the lineage analysis presented in Figure 4. We believe that the inclusion of Cluster 1 in Figure 2 would likely misconstrue the analysis and interpretation of valid immune cell subtypes. Furthermore, with the sequencing data suggesting that cells in Cluster 1 are likely fat body contaminants, there is no reason to include Cluster 1 cells in our hemocyte lineage analysis presented in Figure 4.

2) The definition of Cluster 5 as prohemocytes still relies exclusively on the absence of markers (Figure S10).Raddi et al., (2020) defined two populations of prohemocytes. The authors should at least show the comparison of the markers for cluster and the prohemocytes markers indicated in Raddi et al. report. Also, the presence of Clusters 5 and 7 could reflect the presence of two prohemocyte populations, resembling the situation observed in *Drosophila* (see the work of L Waltzer). At this, point, the interpretation of the data should be more cautious.

We would like to thank the reviewers for this comment. While Raddi et al., (2020) displays two prohemocyte populations in Figure 2C and 2D, there are no further descriptions of these PHem1 and PHem2 populations. In this study, prohemocyte markers are only defined by the HC2 cluster (that includes PHem 1 and PHem 2) in Figure 1, as well as in the supplemental data that define enriched cell markers of each cluster. Frankly, there is no evidence as to how they came to this conclusion, thus limiting any further comparative analysis. However, in our revised manuscript we use these defined markers for HC2 and compare them across our immune cell clusters. Through these comparisons we demonstrate that Cluster 5 (described in our study) most closely resemble the HC2 prohemocyte populations described by Raddi. These comparisons and other markers use to define these populations in Raddi et al., are present in Figure 3—figure supplement 7 of our revised manuscript (lines 275-277 and 374-382).

It should also be noted that like the prohemocyte populations described by Raddi, we see strong expression of NimB2, SPARC, and AGAP004936 in our Cluster 5 cells (Figure 2, Figure 3—figure supplement 7). In both studies, these prohemocyte populations are described as less differentiated cell types and have addressed these similarities between studies at multiple locations of our revised manuscript (lines 275-277, 362-367, and 628-629).

Regarding the potential that there may be further complexity within prohemocyte populations, we would agree. The hierarchical clustering of Cluster 5 cells in Figure 1 supports this possibility, but we have been conservative in the classification of our immune cell identifications without further experimental data. We would also argue that the immature cell populations of Clusters 6 and 7 could also have progenitor functions, yet the expression of several differentiated cell markers of respective granulocyte or oenocytoid populations make them distinct from those of Cluster 5. Hopefully, through the data generated in this manuscript we can develop genetic resources to better study these mosquito immune cell populations to better examine the development, differentiation, and activation of these hemocyte populations in the future. We have summarized this discussion in our revised manuscript (lines 492-495 and 699-705), yet at present resist further speculation without experimental data.

3) Blood-fed vs. naïve conditions.It would help to analyze the data in either naïve or in blood-fed conditions, rather than merging the data from the two conditions. As shown in Figure S3, the eight clusters are represented in very different manner in the two conditions, from 0% Cluster 3 to almost 50% Cluster 2 in blood-fed animals.The interpretation of the impact of blood feeding on the hemocytes (cluster 2 and 4) should be described in the result section, possibly with a figure displaying the differentially expressed genes (present in Table S2).Of note, only Cluster 2 hemocytes are enriched in cell cycle gene: how many genes, in how many cells of the cluster? Can they be preferentially ascribed to a food regimen? Is it possible to compare Cluster 2 to the proliferative cluster observed in Raddi et al.?

We would like to thank the reviewers for this suggestion and have addressed each of the reviewer’s comments in our revised manuscript.

We have added a new figure, Figure 1—figure supplement 4, to highlight some of the genes that show differential gene expression in response to blood-feeding in Clusters 2 and 4, and have added new text in the Results section to address these results (lines 138-147).

In addition, we have also amended the cell cycle analysis now to display the effects of blood-feeding on the entire naïve and blood-fed cell population (Figure 2B), as well as to display the cell cycle analysis within each cluster under both naïve and blood-fed conditions (Figure 2C). The results have also been amended to reflect these changes in the results text in lines 155-163.

We also have added comparisons of our immune cell clusters to those described in Raddi et al. in Figure 2—figure supplement 2 and Figure 3—figure supplement 7, as well as additional text in the discussion of our revised manuscript (lines 533-535 and 628-660).

Moreover, we have also performed our Monocle3 trajectory analysis on naïve, blood-fed, and combined cell populations (Figure 4) to also address potential differences in physiological conditions on cell trajectories and have updated the revised text to reflect these changes (lines 419-422, 427-431, and 543-547).

4) A resource manuscript should provide useful information to the community. This manuscript would gain from a more systematic comparison with the data already available in the literature. For example, Raddi et al., define PPO 4 and 9 as being characteristic of oenocytoids.

We would like to thank the reviewers for this suggestion. In our revised manuscript, we have added several new supplemental figures (Figure 3- supplemental figures 5, 6, and 7) to address these comparisons of our study to previous hemocyte single cell studies in both *Drosophila* and *Anopheles*. We believe that this has been an important exercise, highlighting the similarities and differences of our study with previous published work. This has also forced us to take a highly in-depth look at the studies of Tattikota et al., (*Drosophila*) and Raddi et al., (*Anopheles*), which has significantly improved our manuscript.

Importantly, this comparative analysis demonstrates a severe fault of the Raddi et al., study, in which we question their characterization of oenocytoid populations and have significant doubts that they have actually described populations of oenocytoids. The PPO4 and PPO9 markers used to define “oenocytoid” populations in Raddi et al., while expressed in our oenocytoid cells, are actually enriched in granulocyte populations. These observations are supported by functional data that demonstrates a reduction of PPO4 and PPO9 following phagocyte depletion (Kwon and Smith, 2019- PNAS), and the integral roles of other PPOs in oenocytoid immune function (Kwon et al., 2020-bioRxiv). This is further supported by comparisons to *Drosophila* single cell studies (Tattitkota et al., 2020- *eLife*) which highlight the shared role of PPO1 and lozenge in oenocytoid/crystal cell populations described in our study that are notably absent in the study by Raddi et al. (Figure 3—figure supplement 5). In our revised manuscript, we take a strong, yet tactful response to these discrepancies in the results (lines 358-390) and discussion text (lines 628-636).

5) Could the authors explain why there are no cells from the blood-fed condition in Gate 4?

We would like to thank the reviewers for this comment. This is unfortunately the result of not having enough space on the sample plate to include samples from naïve and blood-fed condition under all of the gating conditions. Blood-fed cells collected under the Gate 4 conditions were originally isolated on another plate, but were never processed. Additional text has been added to the result section (lines 119-120) and within the Figure 1—figure supplement 3 (formerly Figure S3) legend to address this comment in the revised manuscript.

6) On DRAQ5 labelling used for the FACS sorting step, in Figure 1A, several levels of DRAQ5 are observed in the dot plots. Other publications mention the use of DRAQ5 to estimate the ploidy of the cells. Is this what we observe on the dotplots in Figure 1A, the first three gates separating cells based their ploidy? Is this known for mosquitoes' hemocytes? How do the authors interpret this, since cluster 8 is exclusively found in gate 1?

We would like to thank the reviewers for this suggestion. It was an omission in our previous draft not to mention the role of DNA content or ploidy in our FACS gating strategy. There is evidence that mosquito hemocyte populations differ in their ploidy levels (Bryant and Michel, 2014, 2016), although these have not been previously attributed to individual hemocyte subtypes. In our revised manuscript, we have addressed the role of ploidy in our FACS hemocyte collection strategy (lines 127-133).

We have also incorporated a discussion of cell ploidy levels to provide further support that our Cluster 1 cell are likely fat body contaminants, since fat body cells have exhibited polyploidy in mosquitoes and other insects (lines 192-197).

Regarding the isolation of our oenocytoid populations (Clusters 7 and 8) predominantly in Gate 1, the FACS gating would suggest that these cells also have a higher ploidy. At present it is unclear what role this may have. Ploidy levels in immune cell populations haven’t been directly examined before, but it has been suggested that polyploidy may enable cells to quickly respond to physiological stimuli. Given the role of oenocytoids to lyse rapidly upon immune challenge, we believe this is a valid hypothesis that requires further study. We have addressed this in the revised manuscript on lines 127-133, 353-357, and 679-680.

7) line 1048: "higher levels".

We would like to thank the reviewers for catching this typo. This has been corrected in the revised manuscript.

8) Figure 2E: "Ninjurin" is Ninjirin in the text.

We would like to thank the reviewers for catching this typo. This has been corrected in the revised manuscript.

9) Figure S14: Are the data normalised by column or are these expression levels? The unit of the colour gradient should be mentioned (z-score, expression levels?).

We would like to thank the reviewers for this comment. We have added additional details to our experimental methodology (lines 785-787) as well as the individual figure legends for Figure 3—figure supplement 3 (previously Figure S14) as well as the figure legends of other supplemental figures that display similar heat maps (Figure 2—figure supplements 3-6).

10) I would appreciate a somehow more detailed comparison of their hemocyte categories (including cluster 1) with those from Raddi et al. This could be proposed as a supplementary figure for instance. Is there a 1 to 1 correlation between the categories? Also, please indicate discrepancies when relevant, e.g. PPO4 is used as a marker for oenocytoids in Raddi et al. while this gene seems to be expressed in all hemocyte subgroups, and especially in granulocytes, in Kwon et al.

We would like to thank the reviewers for this comment. As in our responses to previous comments (Reviewer comment #4), we have taken extensive efforts in our revised manuscript to make comparisons to Raddi et al. This includes the comparison of Cluster 1 to the fat body and muscle cell contaminants in Figure 2—figure supplement 2, as well as the correlations of Clusters 2-8 to those of Raddi et al., in Figure 3—figure supplement 7. From this analysis, there are not perfect 1:1 correlations between Raddi et al., and that of our own study. While we see homology within prohemocyte and granulocyte populations, there is a stark contrast between oenocytoid populations described between these studies (Figure 3—figure supplements 5 and 6), and we do not believe that the LRR8low/PPO4high oenocytoid populations described by Raddi et al., are in fact oenocytoids. Raddi et al., did not perform any functional assays (such as phagocytosis assays) and rely only on cellular morphology for these identifications. Based on the expression profiles of these LRR8low/PPO4high cells, they more closely resemble granulocytes (likely Cluster 4) in our study. In addition to the information provided in our revised manuscript, complementary studies (Kwon and Smith, 2019- PNAS; Kwon et al., 2020-bioRxiv) further support these conclusions. In addition to the new supplementary figures, we also address this more in-depth comparative analysis at several locations in the revised manuscript (lines 349-377, 389-395, and 608-617).

11) The discussion paragraph where the authors compare their work with the two other scRNAseq reports (l519-544) is somehow awkward. While I do understand the need to provide an overview of the three studies, I would rather insist there (1) on the reason why they managed to recover as many clusters as Raddi et al., despite sequencing fewer cells (strategies not compared in the text), (2) on a more precise comparison of the hemocyte clusters from the 3 studies (see previous point), and (3) summarising current knowledge on the functional characterisation of the different hemocyte categories.

We would like to thank the reviewers for these suggestions. We have extensively rewritten the paragraph in question to reflect many of the changes to the reviewers’ comments above.

In our revised manuscript, we have added a new paragraph (to address point 1 above) to the discussion to describe methodological differences between our study and that of Raddi et al. (lines 627-641).

We have also added new supplementary figures (Figure 2—figure supplement 2, Figure 3—figure supplements 5,6, and 7) and text (lines 608-629) to provide a more in-depth comparative analysis of the different hemocyte subtypes (to address point 2 above) that have described in mosquito hemocyte single-cell studies (Severo et al., 2018; Raddi et al., 2020). However, since Severo et al. only examined 24 cells and described only two hemocyte subtypes, we are limited by the type of comparisons that can be made to this study.

Regarding the reviewers’ final point to summarize current knowledge of the different hemocytes, we have added additional text to the results (lines 358-408) and discussion (lines 492-495, 499-503 and 628-648) to enhance our description of the previous characterization of hemocyte categories. However, we have tried to maintain the balance of our manuscript as being a primary research article. There is obvious need to review the recent advances in hemocyte biology as a result of the recent hemocyte single-cell studies and other recent research advances, but we feel that these are better suited for an independent review article that we hope to address once this study has been published.

12) Changes in cluster populations upon blood feeding (e.g. disappearance of all hemocytes from cluster 3): this could be due to rewiring/changes in gene expression in some specific cells as suggested by the authors. Another explanation could be a change in adherence: if certain cells become sessile after blood feeding, they will not be recovered during mosquito perfusion. Of note, it is unclear why there are no hemocytes identified in gate 4 after blood feeding. As this gating is not selective, one would have expected to recover cells there in both sugar fed and blood fed conditions.

We would like to thank the reviewers for their comments. We have added additional text to both the results (lines 427-431) and discussion (lines 542-547) sections to address the suggestion that changes to cell adherence may also influence our cell populations under different physiological conditions.

As previously mentioned (Reviewer comment #5), the absence of hemocytes in the Gate 4 is an unfortunate artifact of our cell isolations and sample processing. We did not have enough wells on our sample plates to include all of the FACS-gated samples from naïve and blood-fed conditions samples on the same plate. As a result, blood-fed cells collected under the Gate 4 conditions were originally isolated on another plate, but were regrettably never processed. Additional text has been added to the result section (lines 119-120) and within the Figure 1—figure supplement 3 legend to address this comment in the revised manuscript.

13) Figure S1: the cluster color code is different from that of Figure 1. Please use the same for all figures.

We would like to thank the reviewers for this suggestion. We have made these corrections in our revised manuscript.